# How to set AdamW's weight decay as you scale model and dataset size

Xi Wang [1]   Laurence Aitchison [2]

## Abstract

The scaling of the optimal AdamW weight decay hyperparameter with model and dataset size is critical as we seek to build larger models, but is poorly understood. We show that weights learned by AdamW can be understood as an exponential moving average (EMA) of recent updates. This gives critical insights for how to set the weight decay in AdamW, and how the weight decay should scale with model and dataset size. In particular, the key hyperparameter for an exponential moving average is the EMA timescale. Intuitively, the EMA timescale can be understood as the number of recent iterations the EMA averages over. We find that the optimal timescale, measured in epochs, is roughly constant as we change model and dataset size. Moreover, given a learning rate, there is a one-to-one mapping from the EMA timescale to the weight decay hyperparameter. Thus, if the optimal EMA timescale is constant, that implies that as the dataset size increases, the optimal weight decay should fall and as the model size increases, the optimal weight decay should increase (if we follow the muP recommendation for scaling the learning rate). We validate these scaling rules on ResNet-18 and Vision Transformers trained on CIFAR-10 and ImageNet, and on NanoGPT pre-training on OpenWebText. Finally, we found that as training progresses, muP's learning rate scaling breaks down for AdamW unless weight decay is scaled appropriately.

## 1. Introduction

A common machine learning workflow is to prototype by training many smaller models, then at the end do one final training run, with the largest possible model on the largest possible dataset. This workflow is used in many settings,

---

[1] Johns Hopkins University, US [2] University of Bristol, UK. Correspondence to: Xi Wang <xidulu@gmail.com>.

*Proceedings of the $42^{nd}$ International Conference on Machine Learning*, Vancouver, Canada. PMLR 267, 2025. Copyright 2025 by the author(s).

from small-scale student projects to the largest LLM training runs. However, for this workflow to be effective, we need to understand how to transfer optimal hyperparameters from smaller-scale prototyping runs to the final, largest model and dataset. This is very well studied for the optimal learning rate as you scale up width and depth (e.g. Yang et al., 2022; Bordelon et al., 2024; Noci et al., 2024; Everett et al., 2024). However, the largest LLM pretraining runs (e.g. Zhang et al., 2022; Touvron et al., 2023a;b; Tow et al., 2023) use an optimizer with a weight decay term such as AdamW (Loshchilov & Hutter, 2018), and it is not understood how to scale the AdamW weight decay hyperparameter with model sizes, and how to scale hyperparameters as we scale dataset size. Here, we fix this gap, showing how the optimal weight decay transfers across model width and dataset size.

To understand how to transfer weight decay across model and dataset sizes, we argue that AdamW should be understood as an Exponential Moving Average (EMA). Of course, both Adam and AdamW use EMAs to estimate the average gradient, $m_t$, and average squared gradient, $v_t$. That is not what we are talking about. Instead, we observe that in algorithms with decoupled weight decay (i.e. AdamW but not Adam), the weights themselves are an EMA of recent updates (see Sec. 3.1). The key hyperparameter in an EMA is the EMA timescale, which intuitively describes the number of previous iterations that the EMA averages over, which is given by $\tau_{\text{iter}} = 1/(\eta\lambda)$ (Sec. 3.1). In fact, we prove and empirically verify that under a $\eta$-dependent initialization and a fully-scale invariant network, AdamW's optimization trajectory is controlled solely by the timescale $\tau_{\text{iter}}$ for all combinations of learning rate $\eta$ and weight decay coefficient $\lambda$ (Sec. 3.2).

Perhaps more importantly, the EMA perspective tells us how to set the weight decay parameter. Specifically, we can take into account the dataset and batch sizes to measure the timescale in epochs, i.e. $\tau_{\text{epoch}} = \tau_{\text{iter}}/M$, where $M = N/B$ denotes the number of iterations per epoch in a training dataset of $N$ samples under a batch size of B, which measures the fraction of the dataset that contributes to the current weights through the EMA. We empirically observed that the optimal $\tau_{\text{epoch}}$ is *stable* across model and dataset sizes, which, consequently provides a guideline for scaling weight decay. In particular, under a fixed learning rate schedule, as the dataset size $N$ increases, we find that the

optimal weight decay falls (Sec. 4.1). Moreover, if we follow the usual µP recommendation to decrease the learning rate $\eta$ as the model width increases, we find that the optimal weight decay should *increase* as the model size increases (Sec. 4.2). We validate both of these predictions in ResNet, vision transformer trained on CIFAR-10 and ImageNet, and NanoGPT pre-training on OpenWebText.

In summary, our paper's contributions are:

- The weights generated by AdamW can be written as an EMA of recent weight updates.
- The optimal EMA timescale changes little as we scale the model and dataset size.
- The optimal weight decay scales with $1$/dataset size under a fixed learning rate schedule.
- The optimal weight decay increases with model width under µP scaling of learning rate.
- When using AdamW with fixed weight decay, µP learning rate scaling breaks down, but proper weight decay scaling restores its effectiveness

## 2. Background

Yang et al. (2022) considered how to transfer the learning rates in SGD and Adam across model sizes. They considered two key desiderata. First, the initial random weights, $W^0 \in \mathbb{R}^{D \times \text{fan\_in}}$, times an input, $x \in \mathbb{R}^{\text{fan\_in}}$, should not grow or shrink as the model changes size. Here, fan_in is the input dimension, and $D$ is the output dimension. We could write this requirement as, $W^0 x \sim 1$, where we use $\sim$ to indicate "scales with". This gives the usual $W^0 \sim 1/\sqrt{\text{fan\_in}}$ scaling of the standard deviation of the initial random weights. At the same time, they required that the change in the outputs, caused by the first weight update, $\Delta W x$, should not grow or shrink as the model changes size; i.e. $\Delta W x \sim 1$. Yang et al. (2022) show that this requirement implies that the learning rate should scale as $1$/fan_in.

The distinction between the $1/\sqrt{\text{fan\_in}}$ scaling of the initial weights vs. the $1$/fan_in scaling of the learning rate might be a bit puzzling. The intuitive reason for this distinction is given in Yang et al. (2022) Appendix J. In short, for the initial weights,

$$y_i = \sum_{j=1}^{\text{fan\_in}} W_{ij}^0 x_j \qquad (1)$$

we know that $y_i \sim \sqrt{\text{fan\_in}} \times W_{ij}^0$, so to ensure that $y_i \sim 1$, we need $W_{ij}^0 \sim 1/\sqrt{\text{fan\_in}}$. However, this square root scaling only arises in very specific circumstances, when each term in the sum, $W_{ij}^0 x_j$, is zero-mean and uncorrelated. These requirements hold for the initial weights, as they are sampled I.I.D from a zero-mean distribution. However, these conditions do not hold for the update (the precise

reason why is complex; see Yang et al., 2022, for details),

$$\Delta y_i = \sum_{j=1}^{\text{fan\_in}} \Delta W_{ij} x_j, \qquad (2)$$

thus, $\Delta y_i \sim \text{fan\_in} \times \Delta W_{ij}$. To ensure that $\Delta y_i \sim 1$, we therefore need $\Delta W_{ij} \sim 1/\text{fan\_in}$. As the Adam updates, $\Delta W_{ij}$ scale with the learning rate, $\eta$, this implies $\eta$ must also scale with $1/\text{fan\_in}$.

To see why Adam updates scale with the learning rate, consider an Adam update,

$$\Delta W_{ij} = \eta \hat{m}_{ij}/\sqrt{\hat{v}_{ij}} \qquad (3)$$

where we have neglected the small $\epsilon$. Here, $\hat{m}_{ij}$ is an EMA estimate of the gradient $g_{ij}$, while $\hat{v}_{ij}$ is an EMA estimate of the expected squared gradient $g_{ij}^2$. Thus, $\hat{m}_{ij} \sim g_{ij}$ and $\hat{v}_{ij} \sim g_{ij}^2$. That implies $\hat{m}_{ij}/\sqrt{\hat{v}_{ij}} \sim 1$, so looking back at Eq. 3, we have $\Delta W_{ij} \sim \eta$. Hence, to get $\Delta W_{ij} \sim 1/\text{fan\_in}$, we need $\eta \sim 1/\text{fan\_in}$, which is the origin of the $1/\text{fan\_in}$ scaling of the learning rates in µP with Adam.

Importantly, as Yang et al. (2022) considers the size of the first updates relative to the random initial weights, it does not give guidance about how to change the weight decay with the model size, as the weight decay only becomes relevant after many learning steps.

## 3. Methods

### 3.1. AdamW as an EMA

An AdamW update for a single parameter at the $t$th iteration can be written as,

$$w_t = (1 - \eta_t \lambda)w_{t-1} - \eta_t \frac{\hat{m}_t}{\sqrt{\hat{v}_t} + \epsilon} \qquad (4)$$

where $w_t$ is a neural network weight, $\eta_t$ is the learning rate (which potentially varies over time due to scheduling), $\lambda$ is the weight decay, $\epsilon$ is a small constant, $\hat{m}_t$ is a bias-corrected EMA estimate of the expected gradient, and $\hat{v}_t$ is a bias-corrected EMA estimate of the expected squared gradient. Notice that here we adopt the parameterization used by major optimization libraries such as `torch.optim`[1], where the decay for $w_{t-1}$ is controlled by the product $\eta_t \lambda$. This parameterization is different from the form suggested in the original AdamW paper (Loshchilov & Hutter, 2018), which we will discuss in detail in the related work section. Further, we follow the usual convention, established by Loshchilov & Hutter (2018) of scheduling the learning rate but not the weight decay.

Now we will show that these weight updates (Eq. 4) can be *approximately* understood as an EMA[2]. Recall that a

---

[1] https://pytorch.org/docs/stable/generated/torch.optim.AdamW.html

[2] For an introduction to EMAs, see en.wikipedia.org/wiki/Exponential_smoothing

generic exponential moving average estimate, $\text{ema}_t$ can be written as,

$$\text{ema}_t = (1 - {}^1/_{\tau_{\text{iter};t}})\,\text{ema}_{t-1} + {}^1/_{\tau_{\text{iter};t}}\,q_t. \qquad (5)$$

Here, $\text{ema}_t$ forms an exponential moving average estimate of $q_t$, where the potentially time-varying EMA timescale is $\tau_{\text{iter};t}$. We take this EMA (Eq. 5) and set:

$$^1/_{\tau_{\text{iter};t}} = \eta_t\lambda \quad \text{ema}_t = w_t \quad q_t = -\frac{1}{\lambda}\frac{\hat{m}_t}{\sqrt{\hat{v}_t} + \epsilon} \qquad (6)$$

then we recover the AdamW updates (Eq. 4). Of course, AdamW uses EMAs to compute $\hat{m}_t$ and $\hat{v}_t$. Our key insight was that *additionally*, the overall AdamW updates for $w_t$ can themselves be understood as EMAs. Intuitively, the AdamW EMA computes the average of $q_t$, i.e. the noisy minibatch update scaled by the weight decay.

To build intuition, we consider a *fixed timescale*, $\tau_{\text{iter}}$, and write the EMA as a *weighted average* of all minibatch updates (derivation in Appendix B),

$$w_t = {}^1/_{\tau_{\text{iter}}}\sum_{t'=1}^{t} e^{-(t-t')/\tau_{\text{iter}}}\left(-\frac{1}{\lambda}\frac{\hat{m}_{t'}}{\sqrt{\hat{v}_{t'}} + \epsilon}\right). \qquad (7)$$

Thus, the unnormalized weights are $e^{-(t-t')/\tau_{\text{iter}}}$. These weights highlight that recent updates (i.e. $t'$ close to $t$) contribute more to the weighted average. Specifically, for recent updates where $t - t'$ is far smaller than $\tau_{\text{iter}}$, the weights, $e^{-(t-t')/\tau_{\text{iter}}}$, are around 1. At the same time, updates from further into the past contribute less: when $t - t'$ is far larger than $\tau_{\text{iter}}$, then the weights, $e^{-(t-t')/\tau_{\text{iter}}}$, decay to zero. Thus, intuitively, the EMA *averages over roughly the last $\tau_{\text{iter}}$ minibatch updates*.

Notice that AdamW's optimization process differs slightly from a standard EMA process. Standard EMA typically assumes the updates $q_t$'s are independent, while the $q_t$'s (Eq. 6) in AdamW's EMA view are correlated since later updates depend on the weight $\text{ema}_t$, which is affected by the current update $q_t$. However, we can still use EMA as an approximation, and it provides useful insight and intuition for understanding the hyperparameters.

### 3.2. Formalizing intuition from the EMA

In an EMA, perhaps the key parameter is the timescale, $\eta_t\lambda$. Additionally, the initial value is also important; in AdamW, this corresponds to the size of the first update relative to the scale of the initialization, $\eta_0/\sigma$. Indeed, Theorem 1 shows that, under a scale-invariant network, if these two quantities are fixed, the whole optimization trajectory is determined.

**Theorem 1.** *Consider two AdamW optimizers with different learning rates, weight decays, initialization scales and epsilons, $(\eta_t, \lambda, \sigma, \epsilon$ vs. $\eta'_t, \lambda', \sigma', \epsilon')$. Take $w_t$ to be the*

*parameters learned by the first optimizer after the $t$th optimization step, and $w'_t$ to be the parameters learned by the second optimizer. These optimizers are initialized by scaling the same random init,*

$$w_0 = \sigma\xi \qquad\qquad w'_0 = \sigma'\xi, \qquad (8)$$

*where $\xi$ is random noise (e.g. IID Gaussian). Consider a scale-invariant network, in the sense that multiplying the weights, $w$, by an arbitrary positive constant, $1/c > 0$ gives the same output for all inputs, $x$,*

$$\text{net}(x; w) = \text{net}(x; \tfrac{1}{c}w). \qquad (9)$$

*We use the same trajectory of EMA timescales, ${}^1/_{\tau_{iter;t}} = \eta_t\lambda = \eta'_t\lambda'$, the same ratio of the first step, relative to the initial parameter scale, $\frac{\eta_0}{\sigma} = \frac{\eta'_0}{\sigma'}$. Now, the entire trajectory of weights for each optimizer is the same (up to a scalar multiplier,) $w'_t = \frac{1}{c}w_t$ where $c = \frac{\sigma}{\sigma'} = \frac{\eta}{\eta'} = \frac{\lambda'}{\lambda} = \frac{\epsilon'}{\epsilon}$.*

Thus, the network outputs are the same at all points along the trajectory, $\text{net}(x; w_t) = \text{net}(x; w'_t)$. For the proof, see Appendix A.1, and for empirical confirmation, see Appendix A.3. Thus, for scale-invariant networks, only two parameters are relevant to the trajectory: the EMA timescale (and its schedule), $\tau_{\text{iter};t}$, and the scale of the initial learning rate relative to the weight init. We would usually expect the influence of the init to diminish over time, leaving $\tau_{\text{iter};t}$ as the key hyperparameter.

## 4. Scaling weight decay across model and dataset sizes

Since timescale is the key hyperparameter in an EMA, we hypothesize that the optimal timescale should be transferrable across model and dataset sizes. In this section, we empirically verify this hypothesis and consider the implications for the optimal weight decay as we increase the training dataset size (Eq. 11) and model size (Eq. 16).

For all experiments, we implemented the model using Py-Torch and performed optimization using its AdamW implementation. For all tasks, we did not use weight decay on the normalization layers. Experiments were conducted on an internal cluster of Nvidia TitanX/1080ti/2080ti/L40s GPUs, and only one GPU card was used at a time for each run. The results for image classification tasks are the averages from three distinct random seeds, for LLM pre-training tasks we only use one seed due to resource constraints. Comprehensive details on the hyperparameter search range and model specifications can be found in Appendix G.

### 4.1. Transferring weight decay across dataset sizes

First, we study the relationship between the optimal $\lambda$ and the training dataset size. Remember that the timescale, $\tau_{\text{iter}}$,

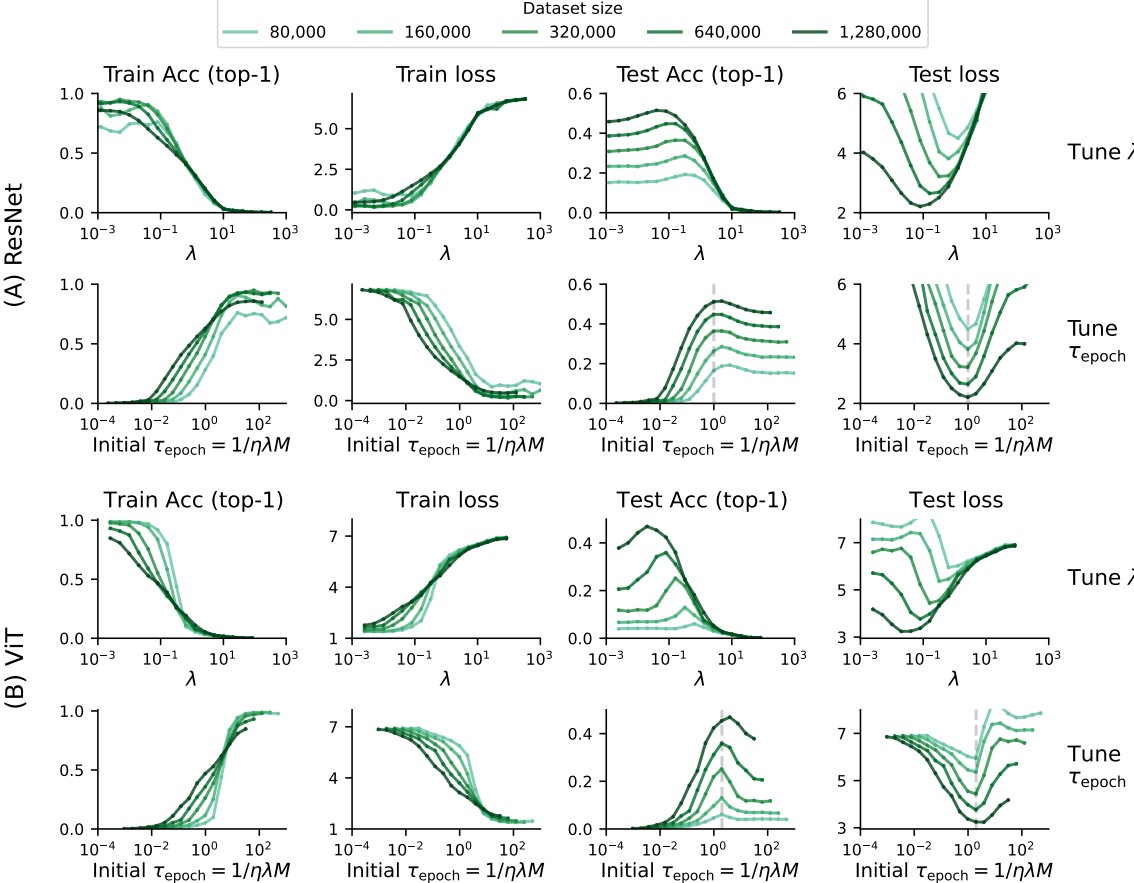

Figure 1: **The optimal $\tau_{\text{epoch}}$ transfers across dataset sizes.** We trained ResNet-18 (A) and ViT (B) on subsets of downsampled ImageNet of various sizes (lines of different colors) under different weight decay (dots on the lines) under a fixed batch size of 100. An initial learning rate of $10^{-3}$ is used with cosine decay scheduling. The performance metrics after 100 epochs are plotted against the weight decay $\lambda$ and the corresponding timescale $\tau_{\text{epoch}}$ computed with the initial learning rate. The dashed lines show the optimal $\tau_{\text{epoch}}$ at a subset size of $320,000$: In both models, the optimal $\tau_{\text{epoch}}$ is fairly stable across dataset sizes whereas the optimal $\lambda$ decreases dramatically as dataset size grows.

intuitively measures the number of previous weight updates that we average over to get the weights. To understand just what proportion of the whole dataset we are averaging over, we propose to work with the timescale in epochs,

$$\tau_{\text{epoch}} = \tau_{\text{iter}}/M = 1/\left(\eta\lambda N/B\right), \qquad (10)$$

where $N$ is the number of training samples, $B$ is the batch size, and $M = N/B$ denotes the number of iterations in an epoch. Therefore, $\tau_{\text{epoch}}$ tells us how many *epochs* of past updates AdamW's EMA averages over, where one epoch passes over all the data in the dataset.

Our prediction is that *as dataset size increases, the optimal $\tau_{\text{epoch}}$ should remain approximately fixed*. If we increase the dataset size with minibatch size fixed, then the number of minibatches in the dataset/an epoch, $M$, increases. Rearranging Eq. (10), that implies that $\tau_{\text{iter}}$ increases with dataset

size, $\tau_{\text{iter}} = M\tau_{\text{epoch}}$. And rearranging $\tau_{\text{iter}} = 1/(\eta\lambda)$,

$$\lambda = 1/\left(\eta\tau_{\text{iter}}\right) = 1/\left(\eta M\tau_{\text{epoch}}\right), \qquad (11)$$

tells us that if the optimal $\tau_{\text{epoch}}$ changes a little as the dataset size increases, we would predict that *the optimal $\lambda$ should decrease as the dataset size increases*. We confirmed this hypothesis on a ResNet-18 and a ViT trained on ImageNet (Fig. 1), along with NanoGPT (124M parameters) trained on OpenWebText (Fig. 2).

For the ImageNet experiments, we used the $32 \times 32$ downscaled version of ImageNet provided by (Chrabaszcz et al., 2017) to ensure that we were able to perform the large number of training runs required to assess performance for different hyperparameter settings and dataset sizes. We trained the models on different subsets of ImageNet, where we randomly drew $80, 160, 320,$ and $640$ samples from each

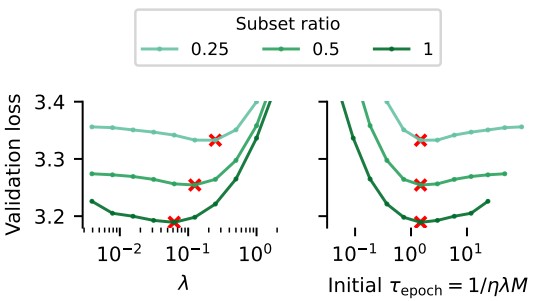

Figure 2: **For LLM pre-training, optimal weight decay shifts with dataset sizes but $\tau_{\text{epoch}}$ transfers.** We trained a 124M NanoGPT on *subsets* of OpenWebText with different sizes (line colors) for 4 epochs under a fixed batch size, an initial learning rate of $6 \times 10^{-4}$ and various $\lambda$ (dots on lines, varied in powers of 2). As dataset size increases, weight decay that gives optimal validation loss (red crosses) decreases, whereas $\tau_{\text{epoch}}$ (Eq. 10) is stable across scales.

of the $1,000$ classes except for the largest subset of $1.28M$ samples in total, where we randomly drew a subset from the whole dataset. We used a fixed batch size of 100 for all runs, an initial learning rate of $10^{-3}$, swept $\lambda$ from $10^{-3}$ to $10^3$, and then plotted the performance metrics after 100 epochs. The top row of Fig. 1A, 1B, shows performance vs. $\lambda$. Note that the optimal $\lambda$ decreases dramatically as the dataset size increases. The bottom row of Fig. 1A, 1B, shows performance metrics vs. $\tau_{\text{epoch}}$. Critically, the optimal $\tau_{\text{epoch}}$ is far more stable than the optimal $\lambda$. Here, we used a cosine schedule which decays to 0.1 of the initial learning rate. In the Appendix, we see similar patterns for a constant schedule, (Fig. 13) and cosine decay to zero (Fig. 14).

Next, we consider training a 12-layer NanoGPT with 124M parameters on subsets of OpenWebText (Gokaslan et al., 2019) of various sizes. We trained the model using the first 1/4, 1/2, and the whole dataset for approximately 4 epochs under a fixed batch size of 122880 tokens with a cosine learning rate decay schedule from $6 \times 10^{-4}$ to $6 \times 10^{-5}$. We swiped $\lambda$ between $2^{-8}$ to 2, and plotted the final validation loss against optimal $\lambda$ and $\tau_{\text{epoch}}$ (Fig. 2), where we again observe that as dataset size grows, optimal $\lambda$ decreases but $\tau_{\text{epoch}}$ is more stable.

Note that in the experiments considered, we used a *fixed* initial learning rate under different dataset sizes. Some recent works (Bjorck et al., 2024; Schaipp et al., 2025) suggest that, under a fixed weight decay and batch size, the optimal learning rate decreases as training run gets longer. We empirically verified this observation in our experiment setting (*top* rows in Fig. 12 A, B in appendix). However, we also notice that if we scale $\lambda_{\text{base}} \propto 1/N$, the optimal $\eta$ becomes consistent across dataset sizes (*bottom* rows in

Fig. 12 A, B in appendix), indicating that scaling $\eta$ may not be necessary if we keep $\tau_{\text{epoch}}$ constant through adjusting $\lambda$ when $N$ scales.

## 4.2. Transferring weight decay across model sizes

Of course, it is also critical to understand how to modify the AdamW hyperparameters as we increase the model size. The most obvious approach is µP (Yang et al., 2022), which predicts that the optimal learning rate should decrease as $1/\text{fan\_in}$, by considering the behavior of the first few learning steps relative to the random initial weights. However, the theory behind µP did not consider how weight decay affects the learned weights in the later phase of the optimization.

Formally, we take $\eta_{\text{base}}$ and $\lambda_{\text{base}}$ as the learning rate and weight decay tuned on a smaller base model, with fan-in of $\text{fan\_in}_{\text{base}}$. Then, we scale the model width by a factor of $s$,

$$s = \text{fan\_in}/\text{fan\_in}_{\text{base}}. \tag{12}$$

The most direct approach using µP scaling with AdamW (e.g. used by Lingle, 2024) scales the learning rate as recommended by µP, while keeping the weight decay fixed,

$$\eta = \eta_{\text{base}}/s \qquad \lambda = \lambda_{\text{base}}, \tag{13}$$

$$\tau_{\text{iter}} = 1/(\eta\lambda) = s/(\eta_{\text{base}}\lambda_{\text{base}}) = s\,\tau_{\text{iter;base}} \tag{14}$$

However, the implied EMA timescale, now changes with model size, while we hypothesize that the optimal timescale should not vary with the model size. Therefore, we conjecture that the scaling in Eq. (14) would break the transferability of optimal learning rate across model sizes if we run the optimization for long enough (see Figure 5AB, top rows).

How do we resolve this issue? We have two desiderata. First, from µP, we should scale $\eta$ with $1/\text{fan\_in}$. Second, we have the timescale $\tau_{\text{iter}} = 1/(\eta\lambda)$ fixed. It might at first seem difficult to reconcile these two desiderata. Indeed, it is impossible in the usual setting where $\lambda$ is fixed. However, it is possible to reconcile them if we allow the weight decay, $\lambda$, to strengthen for larger models.

$$\eta = \eta_{\text{base}}/s \qquad \lambda = s\,\lambda_{\text{base}}, \tag{15}$$

$$\tau_{\text{iter}} = 1/(\eta\lambda) = 1/(\eta_{\text{base}}\lambda_{\text{base}}) = \tau_{\text{iter;base}}. \tag{16}$$

We tested our conjecture that the optimal EMA timescale is constant across model sizes, while the optimal weight decay is not, by training a ResNet-18 of varying widths using µP's codebase on $320,000$ samples subset of downscaled ImageNet, where we randomly drew 320 samples from each class. We scaled the models' widths by factors of 2, giving $s \in \{0.5, 1, 2\}$. We fixed $\eta_{\text{base}} = 10^{-3}$ and used $\eta = \eta_{\text{base}}/s$ to change the learning rate with network size. We swept $\lambda_{\text{base}}$ from $10^{-2}$ to $10^1$. We then considered two ways of modifying the weight decay as we scaled

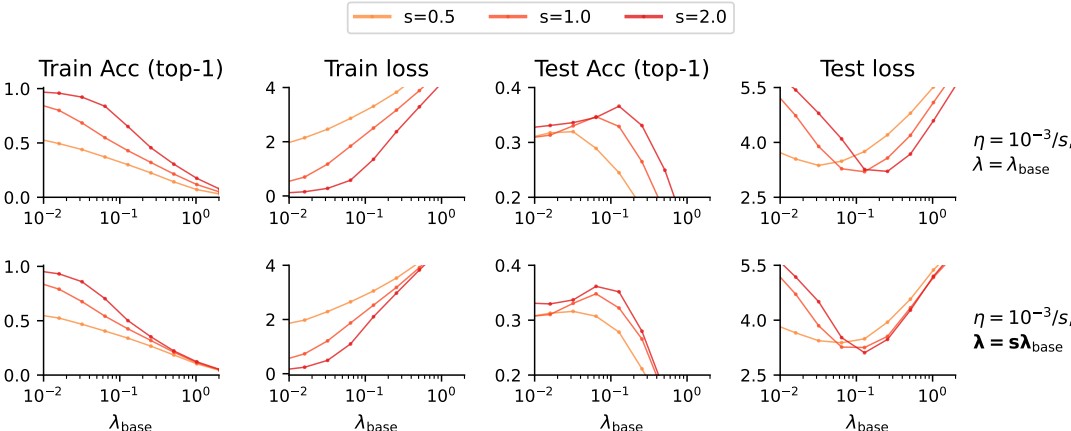

Figure 3: **The optimal $\lambda$ increases with model size whereas the optimal timescale is more stable.** We trained ResNet-18 on a subset of ImageNet 32x32, with varying width factor $s$ (lines of different colors) under a fixed base learning rate $10^{-3}$ with varying weight decay (dots on the lines) and plotted the metrics after 50 epochs vs. weight decay strength. The top row scales the hyperparameters using the direct μP approach (Eq. 13; i.e. fixed $\lambda$), while the bottom row scales the hyperparameters to ensure $\tau_{\text{iter}}$ is fixed (Eq. 15; $\lambda$ increases with model size). Note that as $\eta_{\text{base}} = 10^{-3}$ is fixed, there is a direct relationship between the optimal $\lambda_{\text{base}}$ and the optimal $\tau_{\text{iter;base}}$.

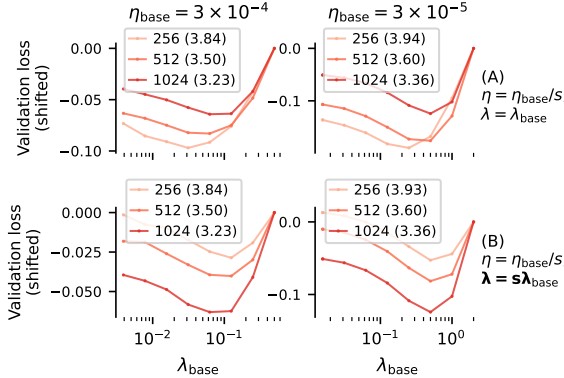

Figure 4: **For LLM pre-training, optimal weight decay increases with model size whereas the optimal timescales transfer.** We trained 8-layer GPTs on OpenWebText, with various widths (line colors), under two $\eta_{\text{base}}$, with actual learning rates for each width scaled following μP with $s = \frac{1024}{\text{width}}$. We plot the final validation loss against various $\lambda_{\text{base}}$ (dots on lines, varied by powers of 2). We align all lines to the rightmost point for better visibility, where the numbers in the brackets denote the actual optimal validation loss at each width. We considered training under two parameterizations, if we keep $\lambda$ decoupled from $s$ then the optimal $\lambda_{\text{base}}$ shows a clear shift with model size (top rows), if $\lambda$ increases with $s$, optimal $\lambda_{\text{base}}$ becomes much stable across widths.

the network and plotted the metrics after 50 epochs against weight decay. Fig. 3 (top row) leaves $\lambda$ fixed as the network size increases (Eq. 13). We can see the optimal $\lambda_{\text{base}}$ varies

dramatically across network sizes. In contrast, Fig. 3 (bottom row) increases the weight decay as the network size increases (Eq. 15), as that leaves $\tau_{\text{iter}}$ unchanged (Eq. 16). It is evident that the optimal $\lambda_{\text{base}}$ and hence $\tau_{\text{iter}}$ is far more stable in this setting. Notably, while e.g. the optimal $\lambda_{\text{base}}$ for test loss on the bottom row does appear different from the others if anything, this would indicate that the relationship between $s$ and $\lambda$ is even stronger than expected. In particular, this point indicates that even as we scale $\lambda$ with $s$, the optimal $\lambda_{\text{base}}$ may still be increasing with $s$. That would seem to indicate that the relationship between $s$ and $\lambda_{\text{base}}$ might be superlinear (e.g. $\lambda_{\text{base}} \propto s^{\alpha}$, where $\alpha > 1$), though we would need more, larger-scale experiments to definitively establish any such super-linearity, which is out-of-scope for the present work.

We conducted similar experiments for LLM pre-training, we considered 8-layer GPTs, with hidden states of widths in $\{256, 512, 1024\}$, achieved by changing the number of attention heads and fixing each head's dimension. We considered two base learning rates, $3 \times 10^{-4}$ and $3 \times 10^{-5}$, trained for 250K iterations, and swiped lambda in the range $2^{-6}$ to 1. The results are shown in Fig. 4, broadly, if we fix $\lambda$ irrespective of $s$ (top row, Eq. 13), optimal $\lambda$ increases with model sizes. However, as we scale $\lambda \propto s$ (bottom row, Eq. 14), the optimal $\lambda_{\text{base}}$ becomes consistent across widths, i.e. confirming that optimal timescale transfers.

Importantly, scaling the weight decay correctly has important implications even for the original μP predictions about how the optimal learning rate transfers across model sizes. Specifically, we trained a ResNet-18 on CIFAR-10

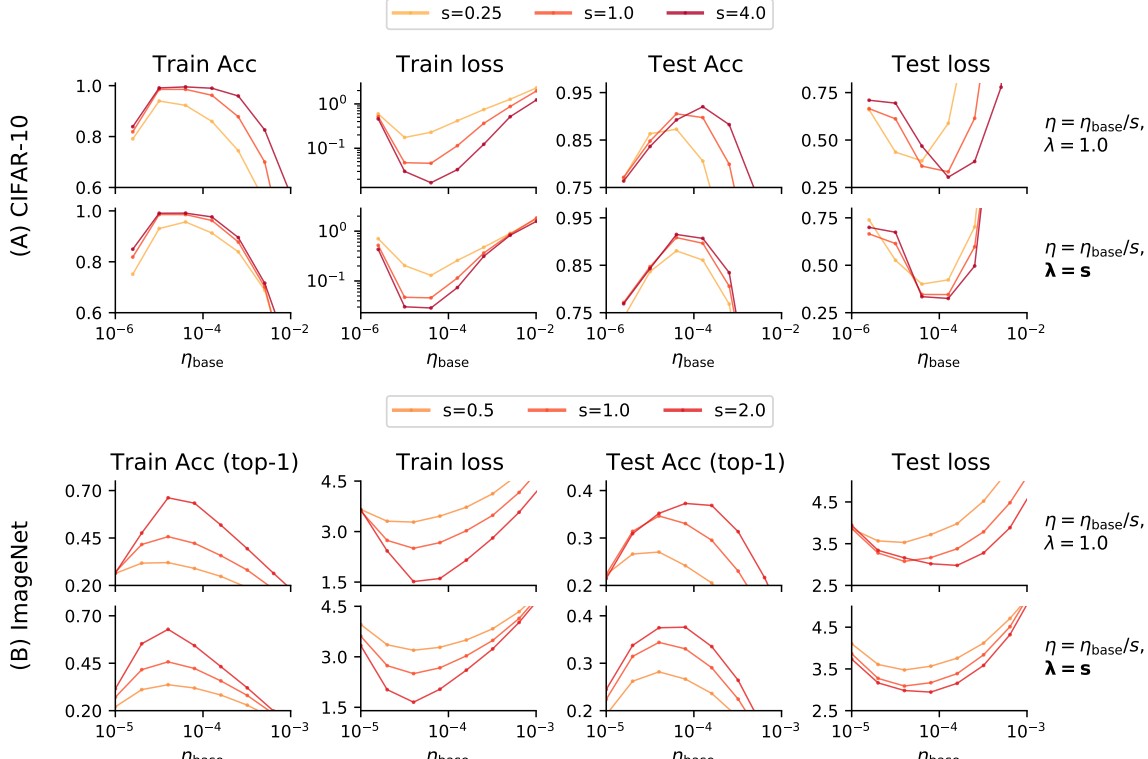

Figure 5: **AdamW breaks the learning rate scaling of μP.** Following the experiment setting in (Yang et al., 2022), we trained a ResNet-18 with varying width factor $s$ (lines of different colors) under various base learning rates $\eta_{\text{base}}$ (x-axis) on CIFAR-10 (A) and a $320,000$ samples subset of ImageNet 32x32 (B). We then plotted the metrics after 200 (for CIFAR-10) and 50 (for ImageNet) epochs against $\eta_{\text{base}}$. The top row scales the hyperparameters using the direct μP approach (Eq. 13; i.e. fixed $\lambda$), while the bottom row scales the hyperparameters to ensure $\tau_{\text{iter}}$ is fixed (Eq. 15; $\lambda$ increases with model size). In both datasets, the direct approach breaks the stability of optimal $\eta_{\text{base}}$ in terms of test metrics due to changing the timescale whereas our scaling allows for consistent $\eta_{\text{base}}$ across model sizes.

and on 320K images downscaled from ImageNet, with a fixed $\lambda_{\text{base}} = 1.0$ and swept $\eta_{\text{base}}$ from $10^{-5}$ to $10^{-2}$. In the top row of Fig. 5A and B, we use the standard μP scaling (Eq. 13), with a fixed weight decay. Remarkably, we found that the optimal base learning rate varied dramatically across model sizes. This indicates that μP scaling of the optimal learning rate breaks down in AdamW. Indeed, the original μP paper (Yang et al., 2022) did not examine AdamW, and this finding is confirmed by recent work on hyperparameter transfer in large-scale transformers (Lingle, 2024; Blake et al., 2024). We hypothesized that this breakdown was due to the timescale changing with model size (Eq. 14), and thus we could restore the usual μP scaling of the learning rate by increasing the weight decay with model size (Eq. 15) to fix the timescale when sweeping $\eta_{\text{base}}$. To confirm this finding, in the bottom row of Fig. 5A and B, we increase the weight decay with model size (Eq. 15). We found that the optimal base learning rate was now far more stable when varying model sizes, confirming our hypothesis. We observed similar behavior when training ViTs of different widths on the

ImageNet subset, the results are presented in Appendix E, and follow-up work, citing us, observed similar behavior in large-scale training of transformers (Blake et al., 2024).

Lastly, we additionally considered constant and cosine decay to zero learning rate schedule for experiments Fig. 3 and Fig. 5, the results are shown in Fig. 16, 17 and Fig. 18, 19, respectively in the appendix.

Finally, the size of the weight decay has implications for the magnitude of the learned weights, and hence whether the outputs $Wx$ remain $\mathcal{O}(1)$ which was one of the key desideratum used to derive μP (Yang et al., 2022). In particular, in AdamW the magnitude of the learned parameters, $W_{ij}$ scales with $1/\lambda$ (or $W_{ij} \sim 1/\lambda$). In our framework, this scaling arises because the quantity that AdamW takes the EMA of (i.e. $q_t$ in Eq.6) scales with $1/\lambda$ (we provide empirical justifications in Appendix. D), but this also arises in Kosson et al. (2023). As we propose $\lambda \sim$ fan_in, this suggests that the learned weights (*not* the initial weights) scale as, $W_{ij} \sim 1/\lambda \sim 1/$fan_in. Thankfully, this is precisely the

scaling we need to ensure that, $y_i = \sum_{j=1}^{\text{fan\_in}} W_{ij} x_j$ does not blow up or shrink as the model size increases (specifically, $y_i \sim 1$). In particular, the learned weights, $W_{ij}$ are a sum of many updates, $\Delta W_{ij}$. Thus, following the same reasoning as that for Eq. (2), we have $y_i \sim W_{ij} \times \text{fan\_in}$. Thus, to ensure that $y_i \sim 1$, we need $W_{ij} \sim 1/\text{fan\_in}$, which is precisely what we get using our proposed scaling.

# 5. Related work

## 5.1. Work after us

There have been a number of follow-up works that follow up on the concepts presented in this paper. First, Blake et al. (2024) confirmed that increasing weight decay as we increase model width fixes learning rate transfer. Second, Bergsma et al. (2025b) has built on our EMA interpretation of AdamW to understand and propose better learning rate schedules. Third, Bjorck et al. (2024) looked at the scaling of optimal learning rates with dataset size. They found that the optimal learning rate decreases with dataset size ($N$), with a fixed batch size, and a fixed weight decay. This broadly what you would expect from our findings, that you need to increase the EMA timescale as the dataset size increases, and you can do that by reducing the weight decay, or reducing the learning rate. However, they found that the dependence of learning rate on dataset size was sublinear. We suspect this is because as you modify the learning rate with weight decay fixed, you simultaneously change two things: the EMA timescale (which you probably do want to change) and the behaviour of the optimizer close to initialization, which you probably do not want to change (Appendix A). Additionally, our experiments verified that $\tau_{\text{epoch}}$ transfers across model widths on small-scale problems while recent works (Dey et al., 2025; Narayan et al., 2025) further conclude that $\tau_{\text{epoch}}$ is also transferrable in *billion-parameter LLM pre-training*.

Lastly, very recent work by Bergsma et al. (2025a) considered two critical problems unstudied in our current manuscript: The effect of batch size as well as the optimal value and transferability of $\tau_{\text{epoch}}$ when you only train the model *for one epoch*, as in LLM pre-training settings. In our experiments, we kept the batch size $B$ fixed when tuning other hyperparameter, however in real world pre-training, batch size $B$ is often tuned extensively to ensure maximum utilization of parallelism, Therefore it is important to understand the how to adjust other hyperparameter as we scale $B$. Fortunately, the heuristic of keeping $\tau_{\text{epoch}}$ (Eq. (10)) constant provides a guidance on how to achieve this: In order to keep $\tau_{\text{epoch}}$ unchanged as $B$ scales, one can have $\eta \propto B$ as suggested by McCandlish et al. (2018), however the picture is often more complicated in practice due to e.g. AdamW's effective learning rate induced by the gradient normalizer (Malladi et al., 2022), as well as

the potential risk of optimization diverging due to learning rate set too large. However, there exists an alternative approach to keep $\tau_{\text{epoch}}$ unchanged as we scale $B$, that is to have $\lambda \propto B$. Indeed Bergsma et al. (2025a) demonstrates the feasibility of such an approach, which shows optimal $\lambda$ scales linearly with respect to $B$, and allows for much more stability and flexibility compared with adjusting $\eta$ when scaling $B$. Additionally, our manuscript only considered high epoch training where the same dataset is repeated multiple times, and suggests that optimal $\tau_{\text{epoch}}$ should stay constant and lies between one and the total number of epochs as dataset sizes increase. However, this may not be the case, LLM pre-training's *one epoch setting*, where each training data point is only presented once. Particularly, Bergsma et al. (2025a) show that optimal $\tau_{\text{epoch}}$ should not stay constant in one epoch setting, but rather follows a power law $\tau_{\text{epoch}}^{\text{optimal}} \approx (\text{TPP})^{-0.527}$, where the optimal $\tau_{\text{epoch}}$ decreases as the token-per-parameter (TPP) increases, and is always smaller than 1 when $\text{TPP} > 1$. Intuitively, this suggests that when the training run is short, it would be better to average over all past updates (high $\tau_{\text{epoch}}$), whereas as training gets longer, it becomes more optimal to forget noisy updates seen in the beginning of the training (low $\tau_{\text{epoch}}$).

## 5.2. Work before us

Further, our EMA view of AdamW provides an explanation for several striking observations made in the literature.

First, the original AdamW paper (Loshchilov & Hutter, 2018) proposed a parameterization in terms of the learning rate, $\eta$ and $\gamma = \eta\lambda$ describing the weight decay. They, as well as a recent blog post (Schaipp, 2024), made an empirical observation that $\eta$ and $\gamma$ were more decoupled than $\eta$ and $\lambda$ (in the sense that the optimal $\eta$ depended strongly on $\lambda$ and less strongly on $\gamma$). We are able to provide a theoretical understanding of their observation, by noting that $\gamma$ is related to our timescale, $\gamma = 1/\tau_{\text{iter}}$. We are therefore able to identify two separate processes one associated with $\eta$ and another associated with $\gamma = 1/\tau_{\text{iter}}$. First, $\eta$ describes the size of the initial updates relative to the random initialization, which is relevant close to initialization (see Appendix A for details). In contrast, $\gamma = 1/\tau_{\text{iter}}$ describes the EMA timescale, which is relevant as the model moves away from the initialization. Finally, Loshchilov & Hutter (2018), did not use these insights to propose how to scale the weight decay with model and dataset size (our key contribution).

Second, our viewpoint (Sec. 3.2) suggests that the effects of the learning rate, $\eta$, on the magnitude of the updates relative to the random initialization should become less relevant than the EMA timescale as training proceeds (Appendix A). While this seems strange (of course the learning rate matters *a lot*), there are some suggestions in the literature that indeed the EMA timescale may be more important in some

settings. Specifically, Wortsman et al. (2024) ran proxies for large-scale LLM pretraining. Fig. 6 in their paper showed that if $\gamma = \eta\lambda = {}^1/\tau_{\text{iter}}$ is fixed, the final validation loss is relatively insensitive to the learning rate, $\eta$. In contrast, if $\lambda$ is fixed, the validation loss shows much more pronounced sensitivity to the learning rate, $\eta$. This makes sense if the key hyperparameter is $\tau_{\text{iter}}$, as modifying the learning rate, $\eta$, with $\lambda$ fixed implies $\tau_{\text{iter}} = 1/(\eta\lambda)$ also changes. However, different from our work, Wortsman et al. (2024) do not explicitly study how $\tau_{\text{iter}}$ transfers across model sizes or if keeping $\lambda$ fixed would break μP. Indeed, the $\gamma = \eta\lambda$ parameterization is often used in SGD, where it is identified as the *effective learning rate* (Zhang et al., 2019b; Li & Arora, 2019; Li et al., 2020; Wan et al., 2021; Li et al., 2022). Recent works extend the effective learning rate perspectives to AdamW by viewing AdamW as sign gradient descent (D'Angelo et al., 2024) or as rotation speed on a fixed radius sphere (Kosson et al., 2023)[3]. However, these literatures do not study how optimal $\eta\lambda$ transfers across model and dataset sizes.

Third, Lingle (2024) assessed the accuracy of μP in large-scale LLM pretraining. Remarkably, they found that the usual μP scaling of the optimal learning rate with model size broke down for AdamW. We provide an explanation and fix for this by noting that μP theory considers only the behaviour of the first few optimization steps, relative to the random initialization. The issue is that if the weight decay, $\lambda$, is fixed, then $\tau_{\text{iter}}$ changes with model size (as $\tau_{\text{iter}}$ depends on $\eta$, and $\eta$ changes with model size following the usual μP recommendations). We propose a fix for the issue by proposing to strengthen weight decay as the model size increases, and validate the fix experimentally.

Fourth, many papers have considered scaling with batch size (which we hold fixed) (Zhang et al., 2019a; Malladi et al., 2022; Wang & Aitchison, 2024). These papers find that when the batch size is small, the optimal learning rate is also small. Then, as the batch size increases, the optimal learning rate also increases. However, there's a limit, the critical batch size, at which point increases in the batch size no longer allow you to increase the learning rate. Once you hit the critical batch size, you no longer get any benefit from increasing the batch size, and you instead must increase the number of iterations (McCandlish et al., 2018; Shallue et al., 2019). Note that while very recent work suggests that the optimal batch size does increase with dataset size, they confirm that this increase is sublinear, implying that the number of training iterations will increase (Zhang et al., 2025). As soon as you need to increase the number of iterations, our recommendation to hold $\tau_{\text{epoch}}$ constant comes into play.

Concurrent work (D'Angelo et al., 2024) suggests that weight decay does not serve as regularization in deep learning, but rather controls minibatch noise, which grows with $\eta\lambda$. Our work provides a similar viewpoint, in the sense that, if you are averaging over more data points (higher $\tau_{\text{iter}} = 1/\eta\lambda$), then the iteration-to-iteration noise will of course be smaller. But the EMA framework also provides you with guidelines for how to set $\lambda$ or equivalently the timescale as you scale model and dataset size.

Of course, μP itself is related work (Yang et al., 2022), and is discussed extensively in the Background section. Van Laarhoven (2017) argue that in Adam, the weight decay hyperparameter changes the scale of the learned weights, and thereby has an influence on the effective learning rate. This effect pops up in our framework in the $1/\lambda$ factor in $q_t$ (Eq. 6), the quantity over which we take the EMA. Additionally, recent work (Busbridge et al., 2023) studies the scenarios where an EMA is used for averaging model weights, such as in self-supervised learning, and proposes hyperparameter scaling rules to achieve the same performance under different batch sizes. Our work differs in that we are interpreting AdamW as EMA over *past updates* rather than using explicitly using EMA for averaging weights. Finally, hints that the optimal weight decay decreases with dataset size have appeared previously in empirical literature when the authors have conducted thorough hyperparameter sweeps, though they did not propose a quantitative, theoretically motivated scaling rule (Zhai et al., 2019).

## 6. Conclusions

We showed that AdamW's weight updates can be understood as an EMA, showed that the optimal EMA timescale is fixed as we scale model and dataset size, and explored the implications for hyperparameter scaling of the weight decay.

## Impact Statement

This paper presents work whose goal is to advance the field of Machine Learning. There are many potential societal consequences of our work, none which we feel must be specifically highlighted here.

---

[3]As pointed out by one of the anonymous reviewers, the timescale parameter in an EMA process can been seen as a measurement of the relative update size studied by Kosson et al. (2023), see Appendix. C for detailed discussion.

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

# A. Confirming the connection between AdamW and weight decay

## A.1. Proof of Theorem 1

Here, we consider two AdamW training trajectories with hyperparameters $\eta_t, \lambda, \epsilon$ and $\eta_t', \lambda', \epsilon'$. We prove that for a specific way of initializing the optimizer state, and if the optimizer hyperparameters are related by,

$$\eta_t' = \tfrac{1}{c}\eta_t \tag{17}$$

$$\lambda' = c\lambda \tag{18}$$

$$\epsilon' = c\epsilon. \tag{19}$$

then the full trajectories are the same. Notice that $1/\tau_{\text{iter};t} = \eta_t\lambda = \eta_t'\lambda'$ remains unchanged with this choice of hyperparameters.

We consider AdamW updates of the form,

$$g_t = \nabla\mathcal{L}(w_{t-1}) \tag{20a}$$

$$m_t = \beta_1 m_{t-1} + (1 - \beta_1)g_t \tag{20b}$$

$$v_t = \beta_2 v_{t-1} + (1 - \beta_2)g_t^2 \tag{20c}$$

$$\hat{m}_t = \frac{m_t}{1 - \beta_1^t} \tag{20d}$$

$$\hat{v}_t = \frac{v_t}{1 - \beta_2^t} \tag{20e}$$

$$w_t = (1 - \lambda\eta_t)w_{t-1} + \eta_t\frac{\hat{m}_t}{\sqrt{\hat{v}_t} + \epsilon} \tag{20f}$$

(And the analogous equations for the primed optimizer). The optimizer state is the variables that persist across timesteps, namely $m$, $v$, and $w$. Specifically, the first optimizer has state $m_t$, $v_t$, and $w_t$ while the second optimizer has state $m_t'$, $v_t'$, and $w_t'$. Our goal is to prove that for a scale-invariant network, as $\lambda$ or the trajectory of $\eta_t$ changes while the trajectory of $1/\tau_{\text{iter};t} = \eta_t\lambda$ is fixed, the states for the different optimizers are related by scale parameters,

$$m_{t-1}' = cm_{t-1}, \tag{21a}$$

$$v_{t-1}' = c^2 v_{t-1}, \tag{21b}$$

$$w_{t-1}' = \tfrac{1}{c}w_{t-1}. \tag{21c}$$

where $c = \lambda'/\lambda = \eta_t/\eta_t'$. The argument proceeds inductively, by assuming that the scaling relations in Eq. (21) hold at timestep $t - 1$, then proving that as a consequence they hold at timestep $t$.

We start by considering the updates for $m$ and $v$, which require the scaling of the gradients. In particular, assuming, that Eq. (21c) holds at timestep $t$, Appendix A.2 tells us that the relationship between gradients is,

$$g_t' = \nabla\mathcal{L}(w_{t-1}') = c\nabla\mathcal{L}(w_{t-1}) = cg_t. \tag{22}$$

Thus, the updates for $m'$ are,

$$m_t' = \beta_1 m_{t-1}' + (1 - \beta_1)g_t'. \tag{23}$$

substituting Eq. (21a) and Eq. (22)

$$m_t' = \beta_1 cm_{t-1} + (1 - \beta_1)cg_t = c(\beta_1 m_{t-1} + (1 - \beta_1)g_t) = cm_t. \tag{24}$$

So the relationship continues to hold for $m$ at timestep $t$. For $v$ updates,

$$v_t' = \beta_2 v_{t-1}' + (1 - \beta_2)(g')_t^2. \tag{25}$$

substituting Eq. (21b) and Eq. (22),

$$v_t' = \beta_2 c^2 v_{t-1} + (1 - \beta_2)cg_t^2 = c^2\left(\beta_2 c^2 v_{t-1} + (1 - \beta_2)g_t^2\right) = c^2 v_t \tag{26}$$

So the relationship continues to hold for $v$ at timestep $t$.

Next, we consider $\hat{m}$ and $\hat{v}$, which are not in themselves state variables, as they can be computed directly from $m$ or $v$ at that timestep.

$$\hat{m}'_t = \frac{1}{1-\beta_1^t}m'_t = \frac{1}{1-\beta_1^t}cm_t = c\hat{m}_t \tag{27a}$$

$$\hat{v}'_t = \frac{1}{1-\beta_1^t}v'_t = \frac{1}{1-\beta_1^t}c^2 v_t = c^2\hat{v}_t. \tag{27b}$$

So the scaling relationships for $\hat{m}$ and $\hat{v}$ are analogous to those for $m$ and $v$.

Finally, we consider the weight updates themselves. The weight updates for the two optimizers are,

$$w_t = (1 - {}^1\!/\!_{\tau_{\text{iter};t}})w_{t-1} + \eta_t\frac{\hat{m}_t}{\sqrt{\hat{v}_t}+\epsilon} \tag{28a}$$

$$w'_t = (1 - {}^1\!/\!_{\tau_{\text{iter};t}})w'_{t-1} + \eta'_t\frac{\hat{m}'_t}{\sqrt{\hat{v}'_t}+\epsilon'} \tag{28b}$$

Substituting $\eta_t = {}^1\!/\!_{\tau_{\text{iter};t}}/\lambda$ and $\eta' = {}^1\!/\!_{\tau_{\text{iter};t}}/\lambda'$,

$$w_t = (1 - {}^1\!/\!_{\tau_{\text{iter};t}})w_{t-1} + {}^1\!/\!_{\tau_{\text{iter};t}}\frac{1}{\lambda}\frac{\hat{m}_t}{\sqrt{\hat{v}_t}+\epsilon} \tag{29a}$$

$$w'_t = (1 - {}^1\!/\!_{\tau_{\text{iter};t}})w'_{t-1} + {}^1\!/\!_{\tau_{\text{iter};t}}\frac{1}{\lambda'}\frac{\hat{m}'_t}{\sqrt{\hat{v}'_t}+\epsilon'} \tag{29b}$$

Now, substituting Eq. (19) and Eq. (18) into the form for $w'_t$,

$$w'_t = (1 - {}^1\!/\!_{\tau_{\text{iter};t}})w'_{t-1} + {}^1\!/\!_{\tau_{\text{iter};t}}\frac{1}{c\lambda}\frac{c\hat{m}'_t}{\sqrt{c^2\hat{v}'_t}+c\epsilon} \tag{30}$$

Substituting Eq. (21c) and Eq. (27),

$$w'_t = (1 - {}^1\!/\!_{\tau_{\text{iter};t}})\frac{1}{c}w_{t-1} + {}^1\!/\!_{\tau_{\text{iter};t}}\frac{1}{c\lambda}\frac{c\hat{m}_t}{\sqrt{c^2\hat{v}_t}+c\epsilon} \tag{31}$$

Cancelling $c$,

$$w'_t = (1 - {}^1\!/\!_{\tau_{\text{iter};t}})\frac{1}{c}w_{t-1} + {}^1\!/\!_{\tau_{\text{iter};t}}\frac{1}{c\lambda}\frac{\hat{m}_t}{\sqrt{\hat{v}_t}} \tag{32}$$

And pulling out $1/c$,

$$w'_t = \tfrac{1}{c}\left((1 - {}^1\!/\!_{\tau_{\text{iter};t}})w_{t-1} + {}^1\!/\!_{\tau_{\text{iter};t}}\frac{1}{\lambda}\frac{\hat{m}_t}{\sqrt{\hat{v}_t}}\right) = \tfrac{1}{c}w_t. \tag{33}$$

As required.

Thus, we have shown that if the scaling relations in Eq. (21) hold at timestep $t-1$, they also hold at timestep $t$. It only remains to complete the inductive proof by showing that the scaling relations hold at initialization. The $m$ and $v$ state variables are usually initialized at zero, $m_0 = v_0 = 0$, and the scaling relations of course hold at zero. However, networks are usually initialized with a fixed parameter scale which does not depend on $\lambda$. To ensure full scale-invariant training, we therefore need to rescale the parameter initializations, to ensure

$$w'_0 = \tfrac{1}{c}w_0 = \tfrac{\eta'}{\eta}w_0. \tag{34}$$

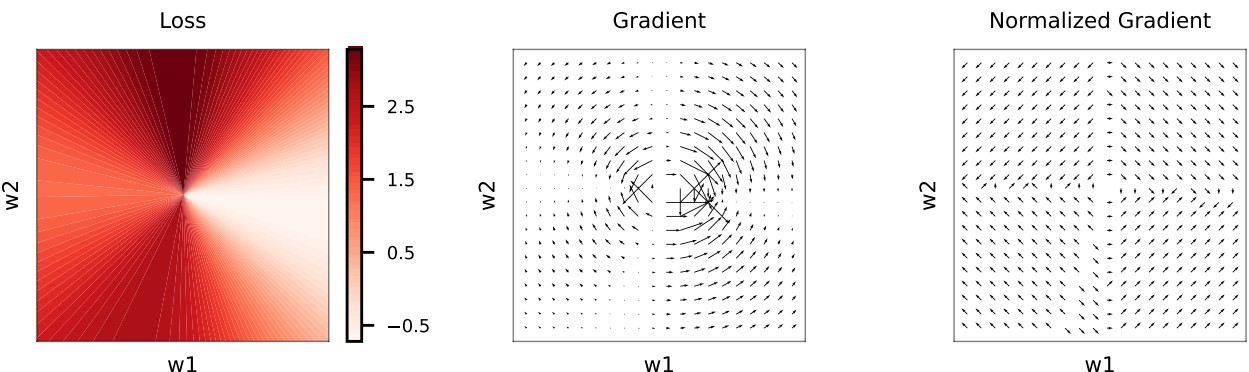

Figure 6: An image of a 2D scale-invariant loss, clearly showing that the gradients get larger closer to the origin.

## A.2. Gradients increase as weights decrease in scale-invariant networks

One important component of our proof is the notion that as weights decrease, the gradient of a scale-invariant loss increases (e.g. see Arora et al., 2019). The geometry is straightforward if you look at an image (Fig. 6). But to spell it out formally, consider a scale-invariant loss,

$$\mathcal{L}(w) = \mathcal{L}(u) \tag{35}$$

where $w = au$. Now, take the partial derivative and use $\frac{\partial u_i}{\partial w_i} = 1/a$,

$$\frac{\partial \mathcal{L}(w)}{\partial w_i} = \frac{\partial \mathcal{L}(u)}{\partial u_i}\frac{\partial u_i}{\partial w_i} = \frac{1}{a}\frac{\partial \mathcal{L}(u)}{\partial u_i} \tag{36}$$

Thus, bigger $w$ (larger $a$) implies smaller gradients wrt $w$.

## A.3. Empirical confirmation of Theorem 1

In this section, we empirically confirm Theorem 1 using a ResNet-18 and a ViT (with QK-layernorm) trained on CIFAR-10. We used AdamW to perform the optimization for 200 epochs with a cosine learning rate schedule to decay the initial learning rate by a factor of 10. We used a fixed batch size of 100 and tested initial learning rates $\eta$ in range $\{10^{-6}, 10^{-5}, \ldots, 10^{-1}\}$. For each $\eta$, we considered $\lambda$ in range $\{10^{-7}/\eta, 10^{-6}/\eta, \ldots, 10^{-2}/\eta\}$, giving us $\tau_{\text{epoch}}$ in range $\{2 \times 10^{-1}, 2 \times 10^0, \ldots 2 \times 10^4\}$ for all $\eta$s.

We begin by looking at the performance of ResNet under different values of $\tau_{\text{epoch}}$ and $\eta$ under the standard configuration. Fig. 7B and Fig. 8B show performance metrics vs. timescale for different learning rates after 200 epochs. We can see from Fig. 7B that the behavior for different learning rates, $\eta$, is very different. In addition, Fig. 8B shows that, while for most $\eta$s, the optimal $\tau_{\text{epoch}}$ in terms of test metrics is the same, the exact performance at the optimal $\tau_{\text{epoch}}$ differs.

This suggests that standard network setups require considerable modification before training trajectories actually become invariant to $\eta$ for a fixed $1/\tau_{\text{iter}}$ (or $\tau_{\text{epoch}}$). In particular, recall that Theorem 1 has two major assumptions about the model: 1. It needs to be scale-invariant (Eq. 9); 2. The initialization needs to be $\eta$-aware (Eq. 34). Therefore, we need to make three key modifications in order for the assumptions to hold:

1. The output weights are not scale-invariant in standard setups, as they are not usually followed by a normalization layer; we therefore introduced a normalization layer after these weights (Appendix A.3.1; row C in Fig. 7, 8)

2. Normalization layers such as batchnorm have scale and bias parameters, which are usually not scale-invariant, as such we adopted a decoupled learning rate for these weights (Appendix A.3.3; row D in Fig. 7, 8).

3. We introduced an initialization that depended on the learning rate, $\eta$ (Appendix A.3.2; row E in Fig. 7, 8).

Combining *all* these modifications together, we found that networks with the same $\tau_{\text{epoch}}$ but different values of $\eta$ demonstrated exactly *the same* learning trajectories (Fig. 7F) and performance metrics (Fig. 8F).

We additionally considered ViT under a similar setting. Notice that ViT needs more extensive modifications to ensure full scale-invariance; full details are presented in Appendix. A.3.4. The performance vs. iteration and timescale for various versions of the model are presented in Fig. 9 and 10. Mirroring the ResNet results, we again found that, under the suggested modifications, the performance becomes invariant to $\eta$ under fixed $\tau_{\text{epoch}}$.

See below for more details of the network modifications required to ensure that experimental results can mirror Theorem 1 in Sec. A.

### A.3.1. OUTPUT-BATCHNORM FOR SCALE-INVARIANT OUTPUT WEIGHTS

Most neural network weights/parameters are scale-invariant. This means that we can multiply the weights/ parameters by an arbitrary constant without changing the network output (Eq. 9). It turns out that most neural network weights/parameters are scale-invariant because they are followed by normalization such as batchnorm or layernorm, which will get rid of any change in scale. However, this is not true for the output layer, as the output layer is not usually followed by a normalization layer.

Therefore, we applied a "global batchnorm" layer after the output layer, i.e., upon the logits. In particular, denote the logits by $Z \in \mathbb{R}^{B \times C}$, where $B$ is the batchsize and $C$ is the number of classes, we first flatten (and reshape) $Z$ into a matrix of shape $BC \times 1$ and then feed the vector into a standard 1D batchnorm, which computes the mean and variance across all $BC$ elements. Finally, we reshape the output of batchnorm back to $B \times C$ and send it to the final softmax layer. The effect of output-batchnorm is shown in row C in Fig. 7 and 8.

### A.3.2. LEARNING-RATE-DEPENDENT INITIALIZATION

Note that Theorem 1 requires the ratio between the initial learning rate and the initial parameter scale to be the same in order for two AdamW optimizers to show the same trajectory. To satisfy the condition in implementation, we fix $\rho = \eta/\sigma = 10^{-3}$, so the initial variance depends on the learning rate, $\sigma = \eta/10^{-3}$.

Confirming this requirement, initializing the network weights in this way seemed to reduce the dependency on $\eta$ for larger values of $\eta$ under fixed $\tau_{\text{epoch}}$ (Fig. 8E).

However, it is worth emphasizing that, $\rho$ itself is another hyperparameter, and different $\rho$ will change the performance (indeed, if we fix the initial variance, then $\rho$ changes as we modify $\eta$).

### A.3.3. DECOUPLING THE LEARNING RATES FOR THE BATCH/LAYERNORM PARAMETERS

In most networks, there are some parameters that are fundamentally non-scale-invariant. These usually include the scale and bias parameters in normalization layers. All of the reasoning in Theorem 1 rely on network output being invariant to scale. So how do we optimize these non-scale-invariant parameters?

It turns out that we already do something different with these parameters. In particular, it is common practice[4] to optimize them using the same learning rate, $\eta$, as the other parameters, but to *drop weight decay*. This standard setting is depicted in Fig. 7E and Fig. 8E, and we can see that $\eta$ does still change the behavior of the optimizer.

The issue is that modifying $\eta$, changes the training trajectory of these non-scale-invariant parameters. The solution is therefore to "decouple" the learning rate for the non-scale-invariant parameters. In particular, we fix the initial learning rate for these non-scale-invariant parameters to $10^{-3}$ regardless of the learning rate $\eta$ for other scale-invariant parameters. Doing so gives Fig. 7F and Fig. 8F, which exhibit almost perfect decoupling, with performance almost entirely independent of $\eta$ for fixed $\tau_{\text{epoch}}$.

### A.3.4. SCALE-INVARIANT ViT

In order to make ViT scale-invariant, we need more extensive modifications beyond simply adding output normalization layer. In particular, we added layernorm after *all* linear layers in the network, which includes

- The embedding layer.

---

[4]We cannot find literature explicitly studying the effect of dropping weight decay for normalization layers' parameters, however, its importance has been noticed in vision tasks, e.g. by Jia et al. (2018) and https://github.com/JiahuiYu/slimmable_networks/issues/15, as well as in language model training: https://github.com/karpathy/minGPT/blob/master/mingpt/model.py#L227

- The query, key and value projection layers.
- The output projection layer after the attention operation.
- The linear layer following each attention block.

## B. The weights in an EMA

Using Eq. (5), and assuming $\text{ema}_0 = 0$,

$$\text{ema}_1 = {}^1\!/_{\tau_{\text{iter}}} q_1 \tag{37}$$

$$\text{ema}_2 = {}^1\!/_{\tau_{\text{iter}}} \left( (1 - {}^1\!/_{\tau_{\text{iter}}}) q_1 + q_2 \right) \tag{38}$$

$$\text{ema}_3 = {}^1\!/_{\tau_{\text{iter}}} \left( (1 - {}^1\!/_{\tau_{\text{iter}}})^2 q_1 + (1 - {}^1\!/_{\tau_{\text{iter}}}) q_2 + q_3 \right) \tag{39}$$

Thus,

$$\text{ema}_t = {}^1\!/_{\tau_{\text{iter}}} \sum_{t'=1}^{t} (1 - {}^1\!/_{\tau_{\text{iter}}})^{t-t'} q_{t'}. \tag{40}$$

In deep learning settings, $\tau_{\text{iter}}$ is much larger than one (implying that we average over many iterations). In that setting, the first-order Taylor expansion is very accurate,

$$1 - {}^1\!/_{\tau_{\text{iter}}} \approx e^{-1/\tau_{\text{iter}}}. \tag{41}$$

Thus,

$$\text{ema}_t \approx {}^1\!/_{\tau_{\text{iter}}} \sum_{t'=1}^{t} (e^{-1/\tau_{\text{iter}}})^{t-t'} q_{t'} = {}^1\!/_{\tau_{\text{iter}}} \sum_{t'=1}^{t} e^{-(t-t')/\tau_{\text{iter}}} q_{t'}. \tag{42}$$

with exact equality as $\tau_{\text{iter}}$ approaches infinity.

## C. Connection between EMA timescale and relative update size

Consider an EMA given by $a_{t+1} = (1 - \gamma)a_t + \gamma b_t$. Assume that the $b_t$s are independently random and identically distributed over time with mean zero, and focus on the relative updates in the steady state (large $t$). The relative update size for the scalar case is defined as

$$r = \sqrt{\frac{\mathbb{E}[(a_{t+1} - a_t)^2]}{\mathbb{E}[a_t^2]}}. \tag{43}$$

To compute $r$, we need to determine $\mathbb{E}[a_t^2]$. We start by expressing $a_t$ in terms of $b_t$ using the EMA formula:

$$a_t = \gamma \sum_{k=0}^{t-1} (1 - \gamma)^k b_{t-1-k} + (1 - \gamma)^t a_0. \tag{44}$$

For simplicity, we assume $a_0 = 0$ in the steady state, leading to:

$$a_t = \gamma \sum_{k=0}^{t-1} (1 - \gamma)^k b_{t-1-k}. \tag{45}$$

Squaring both sides gives:

$$a_t^2 = \gamma^2 \left( \sum_{k=0}^{t-1} (1 - \gamma)^k b_{t-1-k} \right)^2. \tag{46}$$

Expanding the square, we have:

$$a_t^2 = \gamma^2 \left( \sum_{k=0}^{t-1} (1 - \gamma)^{2k} b_{t-1-k}^2 + 2 \sum_{0 \le k < j < t} (1 - \gamma)^{k+j} b_{t-1-k} b_{t-1-j} \right). \tag{47}$$

Due to the independence and zero mean assumption of $b_t$, the cross terms $b_{t-1-k} b_{t-1-j}$ for $k \neq j$ have zero expectation. Thus, we have:

$$\mathbb{E}[a_t^2] = \gamma^2 \sum_{k=0}^{t-1} (1-\gamma)^{2k} \mathbb{E}[b_{t-1-k}^2]. \tag{48}$$

Assuming $\mathbb{E}[b_t^2] = \sigma^2$ and using the geometric series sum formula for large $t$, we get:

$$\mathbb{E}[a_t^2] = \frac{\gamma^2 \sigma^2}{1 - (1-\gamma)^2}. \tag{49}$$

This simplifies to:

$$\mathbb{E}[a_t^2] = \frac{\gamma^2 \sigma^2}{\gamma(2-\gamma)}. \tag{50}$$

Next we will simplify $\mathbb{E}[(a_{t+1} - a_t)^2]$, we begin by expanding the inner term

$$(a_{t+1} - a_t)^2 = \gamma^2 (b_t^2 - 2 b_t a_t + a_t^2). \tag{51}$$

Taking expectations, we have:

$$\mathbb{E}[(a_{t+1} - a_t)^2] = \gamma^2 (\mathbb{E}[b_t^2] - 2\mathbb{E}[b_t a_t] + \mathbb{E}[a_t^2]). \tag{52}$$

Assuming $b_t$ and $a_t$ are independent and $b_t$ has mean zero, $\mathbb{E}[b_t a_t] = 0$. Thus:

$$\mathbb{E}[(a_{t+1} - a_t)^2] = \gamma^2 (\mathbb{E}[b_t^2] + \mathbb{E}[a_t^2]). \tag{53}$$

Using $\mathbb{E}[b_t^2] = \sigma^2$ and substituting $\mathbb{E}[a_t^2]$ from the previous derivation:

$$\mathbb{E}[(a_{t+1} - a_t)^2] = \gamma^2 \left( \sigma^2 + \frac{\gamma^2 \sigma^2}{\gamma(2-\gamma)} \right). \tag{54}$$

This simplifies to:

$$\mathbb{E}[(a_{t+1} - a_t)^2] = \gamma^2 \sigma^2 \left( 1 + \frac{\gamma}{2-\gamma} \right). \tag{55}$$

Finally, simplifying further gives:

$$\mathbb{E}[(a_{t+1} - a_t)^2] = \frac{2\gamma^2 \sigma^2}{2-\gamma}. \tag{56}$$

Thus, the expression for $r$ becomes:

$$r = \sqrt{\frac{2\gamma \sigma^2}{\sigma^2}} = \sqrt{2\gamma}, \tag{57}$$

which indicates that the timescale parameter in the EMA is directly linked with the relative update size.

## D. Weight norm

Notice that the EMA perspective of AdamW (Eq. (6)) suggests that AdamW averages over recent $q_t = -\frac{1}{\lambda} \frac{\hat{m}_t}{\sqrt{\hat{v}_t} + \epsilon}$, and if we assume $\frac{\hat{m}_t}{\sqrt{\hat{v}_t} + \epsilon}$ is of scale $\mathcal{O}(1)$ (Balles & Hennig, 2018), the weights acquired from AdamW should be of magnitude $\mathcal{O}(\frac{1}{\lambda})$.

To verify this, we trained a ResNet on CIFAR-10 for 1000 epochs under a batch size of 100, with an initial timescale $\tau_{\text{iter}} = 1/(\eta_0 \lambda) = 10^5$. We used a cosine learning rate schedule, starting from $\eta_0 = 10^{-5}/\lambda$ and decaying to 0. We then swiped $\lambda$ between $\{10^{-3}, 3 \times 10^{-3}, 10^{-2}, \ldots, 0.3, 1.0\}$ and recorded the *average absolute value* of all dimensions in the convolutional and linear layer weights (i.e. weights where we adopt weight decay) under different $\lambda$.

Fig. 11 left plots the final weight magnitudes against $\lambda$, with both X and Y axes log-scaled. By comparing it with the reference line $y = 0.01/x$ (gray, dashed), we can see the magnitude against $\lambda$ line shares the same slope, indicating that the weight magnitude approximately follows $\mathbb{E}[|W_i|] = c/\lambda$, where $c$ is a constant. Fig. 11 right plots the weight magnitudes throughout optimization, where we can see that all configurations start with the same magnitude at initialization, but gradually converge to our predicted magnitudes as optimization proceeds.

# E. Additional experiment results

## E.1. ViT scaling experiments on ImageNet

We additionally considered training a ViT on the 320K subset of ImageNet 32x32. In particular, we used a ViT with QK-layernorm similar to the one used in the main text, and we varied the width/size of the model by multiplying the number of hidden dimensions and internal linear layers' width with a factor $s$. Similar to the setting in the main text, we tested the width factor in $\{0.5, 1.0, 2.0\}$, used a fixed $\lambda_{\text{base}} = 1.0$, and swept $\eta_{\text{base}}$ in $(2.5 \times 10^{-6}) \times 2^i$ with $i \in \{0, 1, \ldots, 11\}$.

The results are presented in Fig. 15, where we plotted the performance after 50 epochs against $\eta_{\text{base}}$. We again considered the direct μP scaling (Eq. 13; top row) and our proposed scaling (Eq. 15; bottom row), and we can see that the standard scaling breaks the stability of optimal learning rate in terms of test metrics whereas our proposed scaling is much more stable.

# F. Note on muP library

We used the `mup` library[5] from the authors of Yang et al. (2022) for varying-model-size experiments. In particular, for ResNet experiments in Fig. 3 and 5, we directly use the `ResNet` codebase under the `examples` folder together with the provided `MuAdamW` optimizer to run our experiments, as this library takes care of details such as the scaling of the learning rate and initialization for the input/output layers. For ViT experiments in Fig. 15, we manually constructed the required model shape file using the provided `make_base_shapes` function and used the provided `MuReadout` module as the classification head.

Importantly, this library does come with a `decoupled` keyword argument. Given the connections between $\tau_{\text{iter}}$ and the parameterization for AdamW originally proposed in the original "Decoupled Weight Decay Regularization" paper (Loshchilov & Hutter, 2018), you would have thought that you could implement our proposed scalings using `decoupled=True`. However, it turns out that as of writing, to get our proposed scaling for $\lambda$ (Eq. 15), you need to set `decoupled=False`[6]. This may be fixed in the future, but in any case, if using the `mup` library, it is critical to check the `mup` source to see precisely what scalings you are getting for $\lambda$.

# G. Extended experiment setups

## G.1. Model specification

For ResNet-18 experiments, we utilized the implementation from https://github.com/kuangliu/pytorch-cifar/. For both CIFAR-10 and ImageNet, we used random cropping and horizontal flip as data augmentation and we used cross-entropy as the loss function. For μP experiments with ResNet, we use the codebase [7] provided by Yang et al. (2022), which is an adaptation of the codebase provided by the GitHub user `kuangliu`.

For ViT, we adopted the implementation from https://github.com/omihub777/ViT-CIFAR/tree/main. We also incorporated QK layernorm suggested by Dehghani et al. (2023) and Wortsman et al. (2024) in order to stabilize the training when sweeping learning rates. The loss function is again chosen as cross-entropy loss. Additionally, when training on CIFAR-10, we follow the suggestions in the original codebase to use auto augmentation (Cubuk et al., 2019) and label smoothing (Müller et al., 2019) with $\alpha = 0.1$ to alleviate overfitting. We indeed found these techniques crucial for reaching the level of test accuracy reported by the repo. For ImageNet experiments, we used label smoothing with standard data augmentation: random cropping and horizontal flip.

For dataset size scaling experiments with NanoGPT in Fig. 2, we used the configuration code provided by (D'Angelo et al., 2024)[8]. The model structure is identical to the standard 124M version of NanoGPT, which contains 12 layers, each layer has 12 attention heads with a head dimension of 64, i.e. an embedding size of 768. Identical to the original NanoGPT, (D'Angelo et al., 2024) uses a micro-batch size of 12 sequences and 40 gradient accumulation steps at each iteration. However, to reduce memory consumption, it uses a fixed context length of 256 tokens instead of the original 1024-token context length.

---

[5]https://github.com/microsoft/mup
[6]https://github.com/microsoft/mup/blob/19814971934ef91dd546f88e913fc963e096d11c/mup/optim.py#L79
[7]https://github.com/microsoft/mup/tree/main/examples/ResNet
[8]https://github.com/tml-epfl/why-weight-decay/blob/main/large_language_models/config/train_gpt2_small_block256.py

For width transfer experiments in Fig. 4, we used the codebase from the recent project nanoGPT-mup[9]. In particular, we trained transformers with 8 layers, with a fixed attention head dimension of 64, and varied the number of attention heads between $\{4, 8, 16\}$. We additional included QK layernorm for better stability when sweeping hyperparameters. The model is trained with a micro-batch size of 8 sequences, 16 gradient accumulation steps, and a fixed context length of 256 tokens.

### G.2. Hyperparameter range

In Sec. 4.1, for the experiments in Fig. 1, we used $\lambda = (2.5 \times 10^{-6}) \times 2^i$ with $i \in \{0, 1, \ldots, 15\}$.

In Sec. 4.2, for the experiments in Fig. 3, we used $\lambda_{\text{base}} = 10^{-3} \times 2^i$ with $i \in \{0, 1, \ldots, 11\}$.

For the CIFAR-10 experiments in Fig. 5A we used $\eta_{\text{base}} = (2.5 \times 10^{-4}) \times 4^i$ with $i \in \{0, 1, \ldots, 5\}$.

For the ImageNet experiments in Fig. 5B we used $\eta_{\text{base}} = (2.5 \times 10^{-6}) \times 2^i$ with $i \in \{0, 1, \ldots, 11\}$.

For the NanoGPT experiments in Fig. 2, we used $\lambda = 2^i$ with $i \in \{-8, -7, \ldots, 1\}$.

For the NanoGPT experiments in Fig. 4. We used $\lambda_{\text{base}} = 2^i$ with $i \in \{-8, -7, \ldots, -1\}$ for $\eta_{\text{base}} = 3 \times 10^{-4}$ and $\lambda_{\text{base}} = 2^i$ with $i \in \{-6, -5, \ldots, 1\}$ for $\eta_{\text{base}} = 3 \times 10^{-5}$

## H. Licenses

- ResNet-18 from https://github.com/kuangliu/pytorch-cifar/ is MIT licensed.
- ViT from https://github.com/omihub777/ViT-CIFAR/tree/main is MIT licensed.
- CIFAR-10 https://www.cs.toronto.edu/~kriz/cifar.html (No license evident).
- ImageNet license is available at https://www.image-net.org/download.
- The mup library is MIT licensed.
- NanoGPT pre-training code from https://github.com/tml-epfl/why-weight-decay is MIT licensed.
- NanoGPT-mup code from https://github.com/EleutherAI/nanoGPT-mup is MIT licensed.

---

[9] https://github.com/EleutherAI/nanoGPT-mup

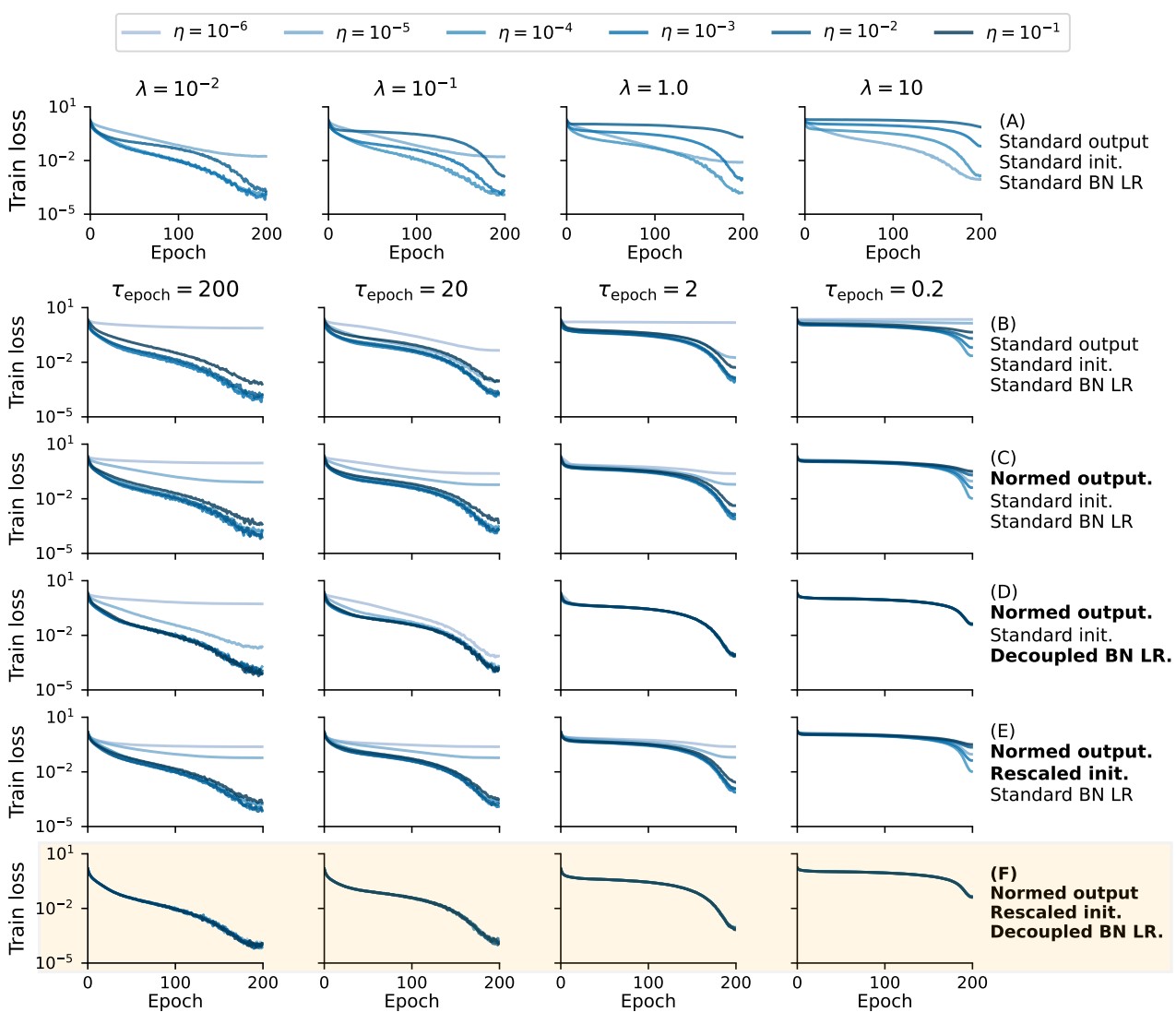

Figure 7: **Proposed modifications decouple the training trajectory from $\eta$ under fixed $\tau_{\textbf{epoch}}$.** We trained a ResNet under various learning rate $\eta$ (lines) and timescale $\tau_{\text{epoch}}$ (columns) and plotted the training loss against epochs. We considered standard vs. normalized output layer (row B vs. row C, Sec. A.3.1); using $\eta$ for batchnorm parameters vs. a separate decoupled learning rate (row C vs. row D, Sec. A.3.3); using standard initialization scale vs. $\eta$-dependent initialization (row C vs. row D, Sec. A.3.2; and lastly, combining all modifications (row F), which allows the trajectory to be independent of $\eta$ under a fixed timescale.

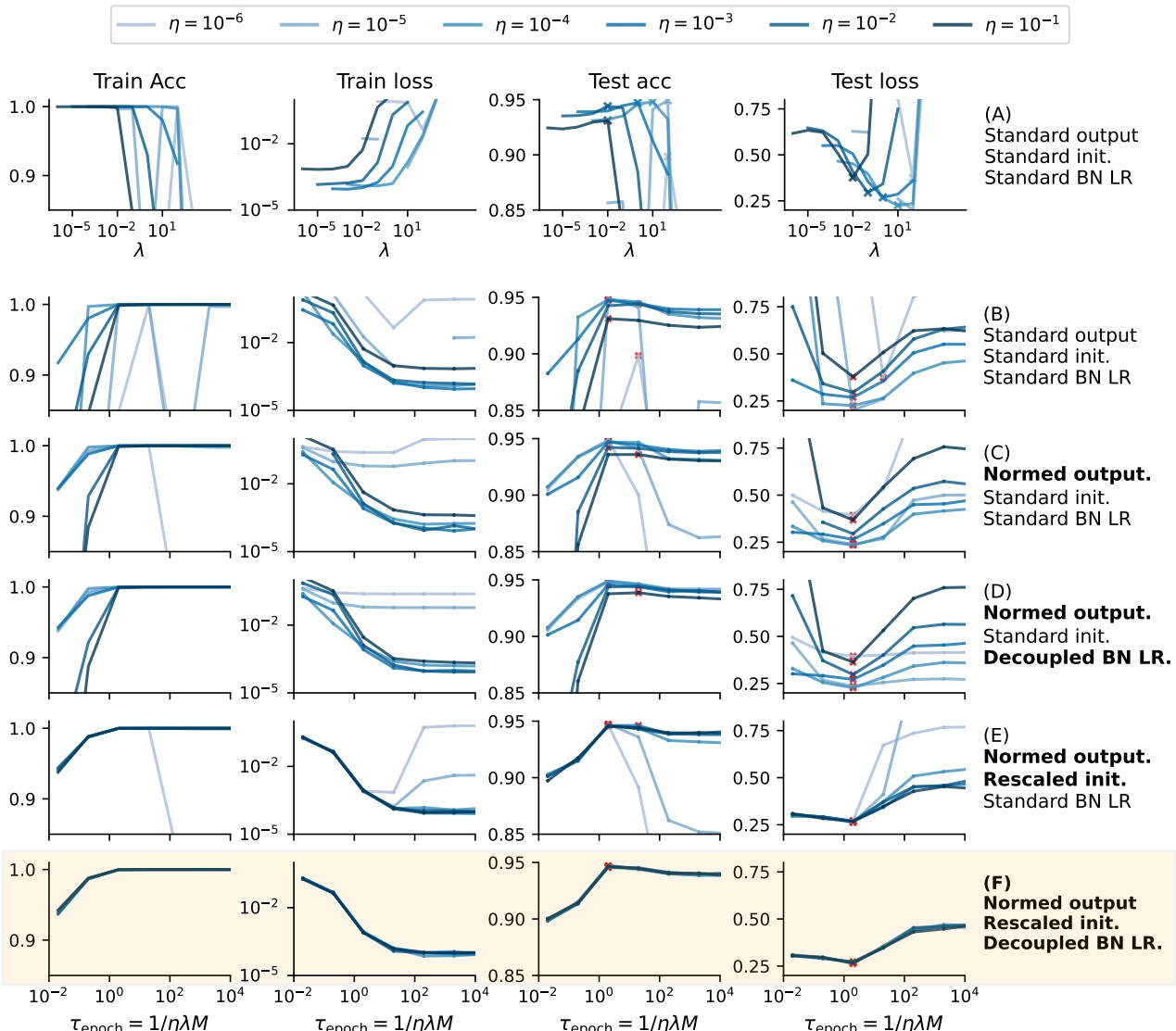

Figure 8: **Under proposed modifications, the performance is only controlled by the timescale.** We trained a ResNet-18 on CIFAR-10 and plot the metrics at the end of the optimization under different initial learning rates (lines) against the timescale (x-axis). Similar to Fig. 7, from row B to F, we considered different levels of modification to the network. From row B to E, while the optimal $\tau_{\text{epoch}}$ (marked by red crosses) lies in close range across different $\eta$, the exact performance still varies by $\eta$. Whereas in row F, after adopting all three modifications, the performance metrics become invariant to $\eta$ under the same $\tau_{\text{epoch}}$.

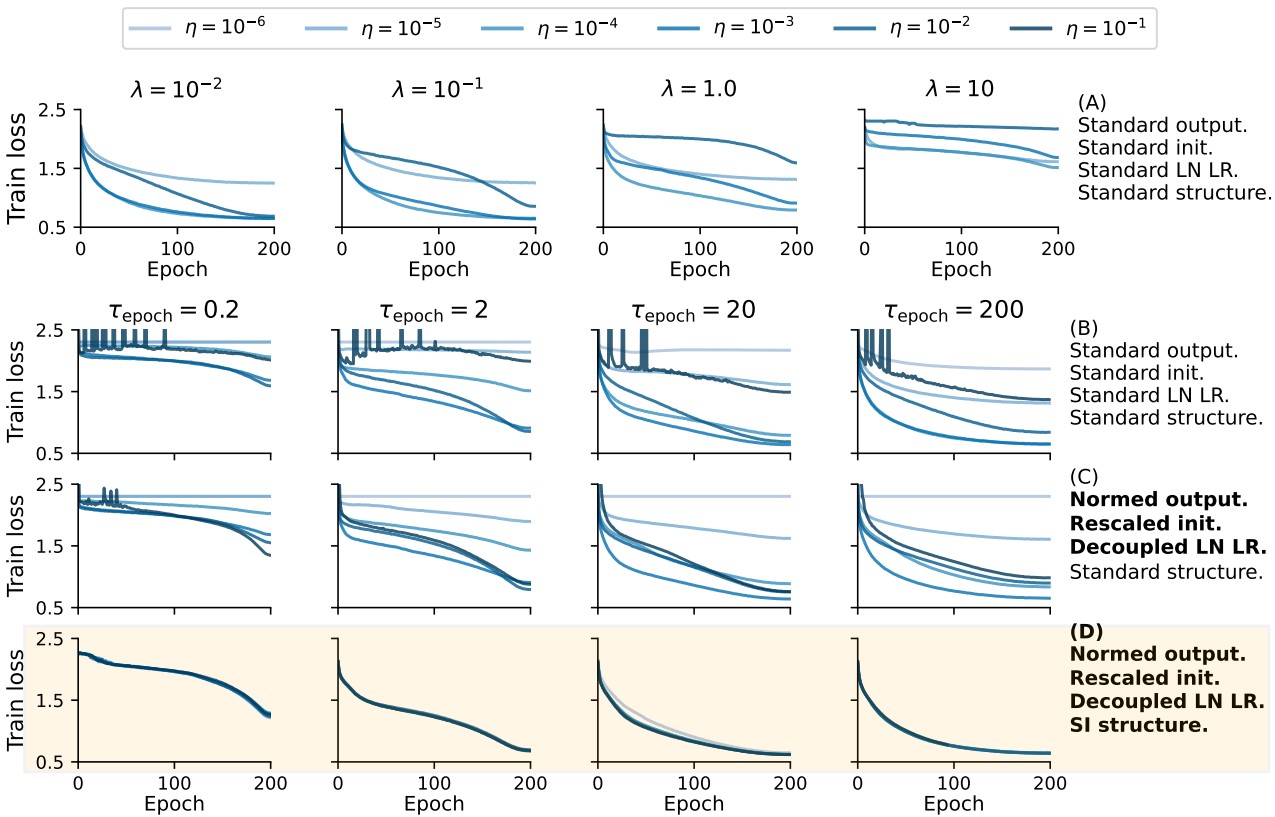

Figure 9: **Proposed scale-invariant structure decouples the training trajectory from $\eta$ under fixed $\tau_{\mathbf{epoch}}$ for ViT.** We trained a ViT under various learning rates $\eta$ (lines) and timescale $\tau_{\text{epoch}}$ (columns) and plotted the training loss against epochs. We first considered standard model vs. model with the three modifications used in ResNet experiments (row B vs. row C, Sec. A.3.1, A.3.3, A.3.2), where the training loss traces still show discrepancy for different $\eta$. We then considered further modification (Sec. A.3.4, row D) to ensure scale-invariance (SI), which successfully decoupled the loss trajectory from $\eta$ under the same $\tau_{\text{epoch}}$.

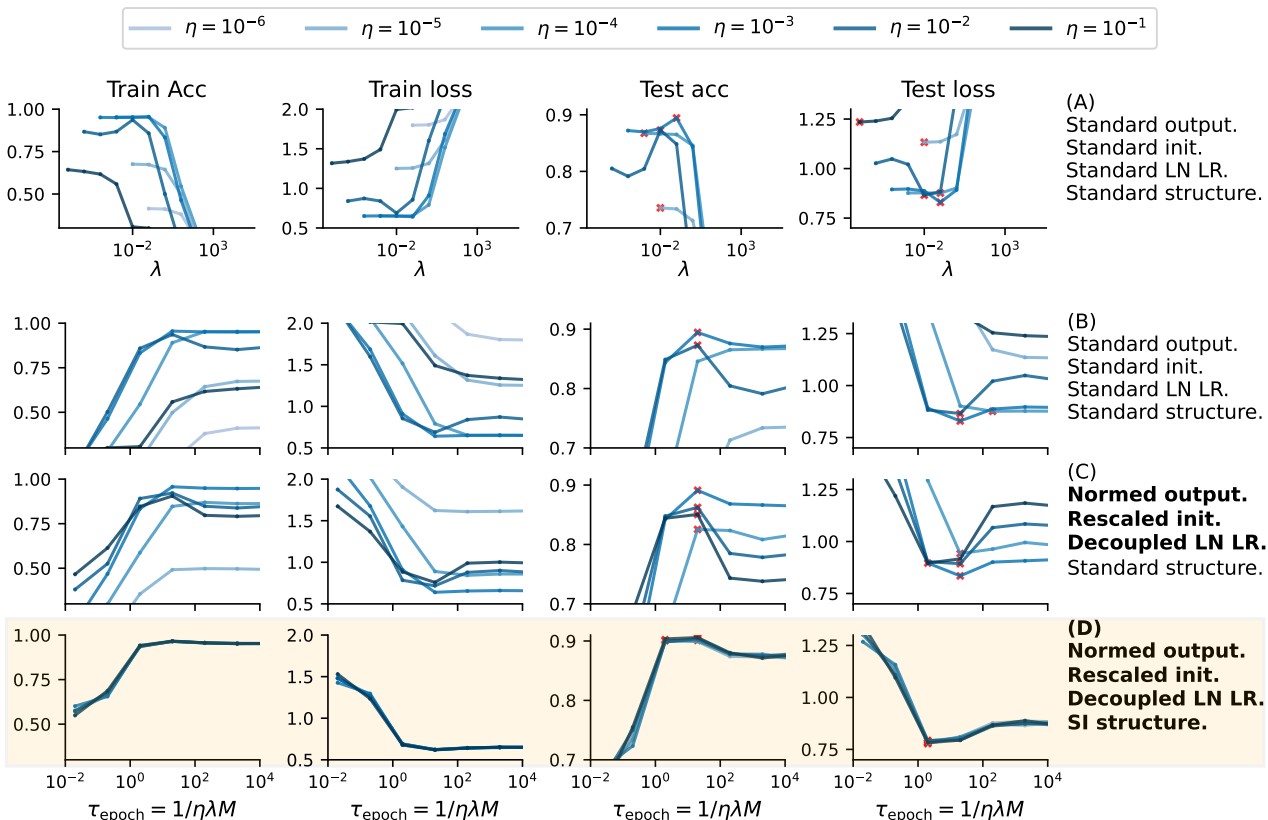

Figure 10: **Under proposed modifications, ViT shows performance controlled only by the timescale and irrelevant to the learning rate.** Similar to Fig. 9 but now we plotted the final performance metrics against $\tau_{\text{epoch}}$ (x-axis) under different $\eta$ (lines). In rows B, and C, the model is not fully scale-invariant and the initialization is not $\eta$-dependent, as such the performance varies between $\eta$s under a fixed $\tau_{\text{epoch}}$. In row D, when all modifications are incorporated, i.e. assumptions in Theorem. 1 are satisfied, the performance becomes only dependent on $\tau_{\text{epoch}}$.

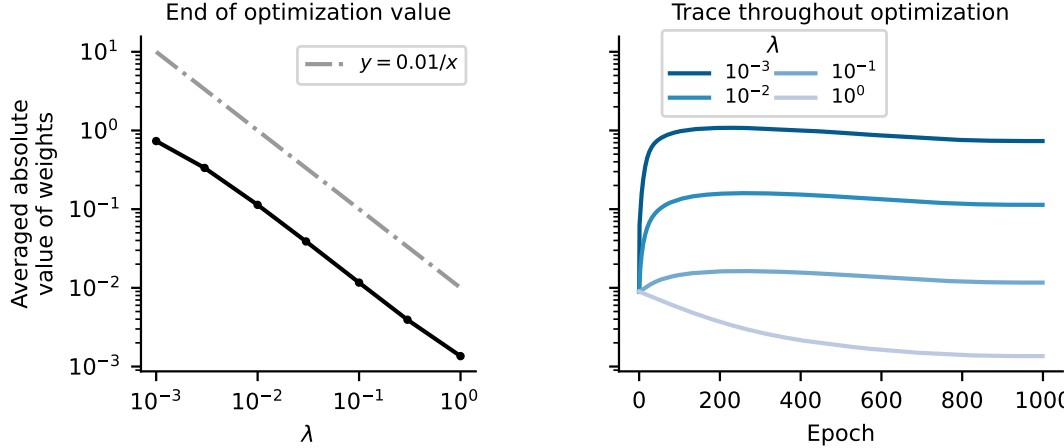

Figure 11: **Weight magnitudes scale inversely with $\lambda$ AdamW optimization**. We trained a ResNet-18 on CIFAR-10 using AdamW with cosine learning rate decay over 1,000 epochs. The weight decay parameter $\lambda$ varies from $10^{-3}$ to 1, with initial learning rate $\eta = 10^{-5}/\lambda$. *Left*: Final average absolute weight values $\mathbb{E}[|W_i|]$ in convolutional and linear layers versus $\lambda$, showing approximate $1/\lambda$ scaling (same slope as the reference dashed gray line). *Right*: Evolution of weight magnitudes throughout training for different $\lambda$ values, different runs start with the same magnitude and gradually converge to $\mathcal{O}(1/\lambda)$.

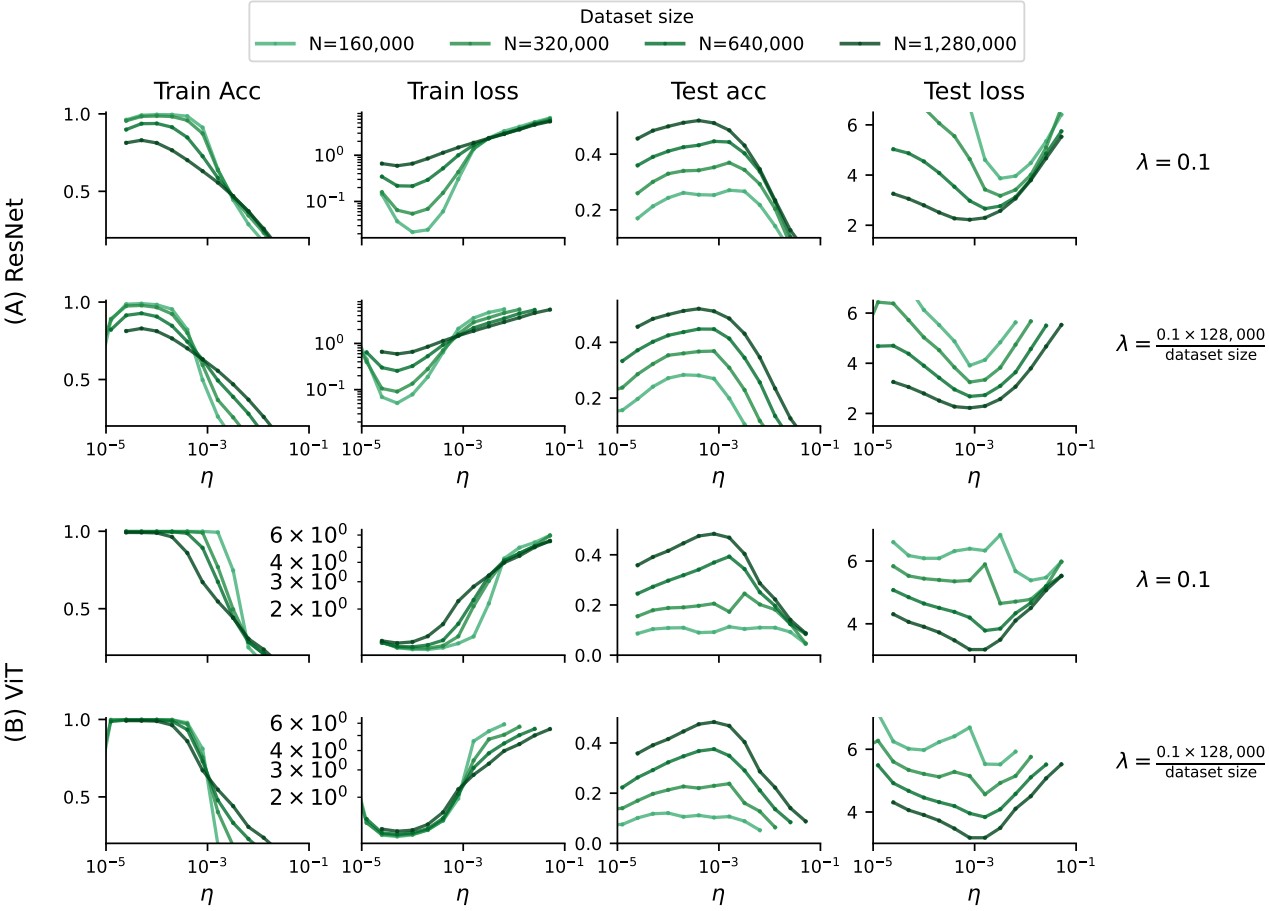

Figure 12: **Under the suggested weight decay scaling, the optimal learning rate is stable across training length.** Mirroring Fig. 1 in the main text, we trained the model for 100 *epochs* with different dataset sizes under a fixed batch size. Using a fixed weight decay (top rows in subfig. A, B), the optimal learning rate *decreases* with the dataset size. Under our suggested weight decay scaling (bottom rows in subfig. A, B), where $\lambda \propto \frac{1}{\text{dataset size}}$, the optimal learning rate becomes more stable across dataset sizes. Note that we select the values for $\lambda$ as 0.1 as they were close-to-optimal for the experiments in Fig. 1.

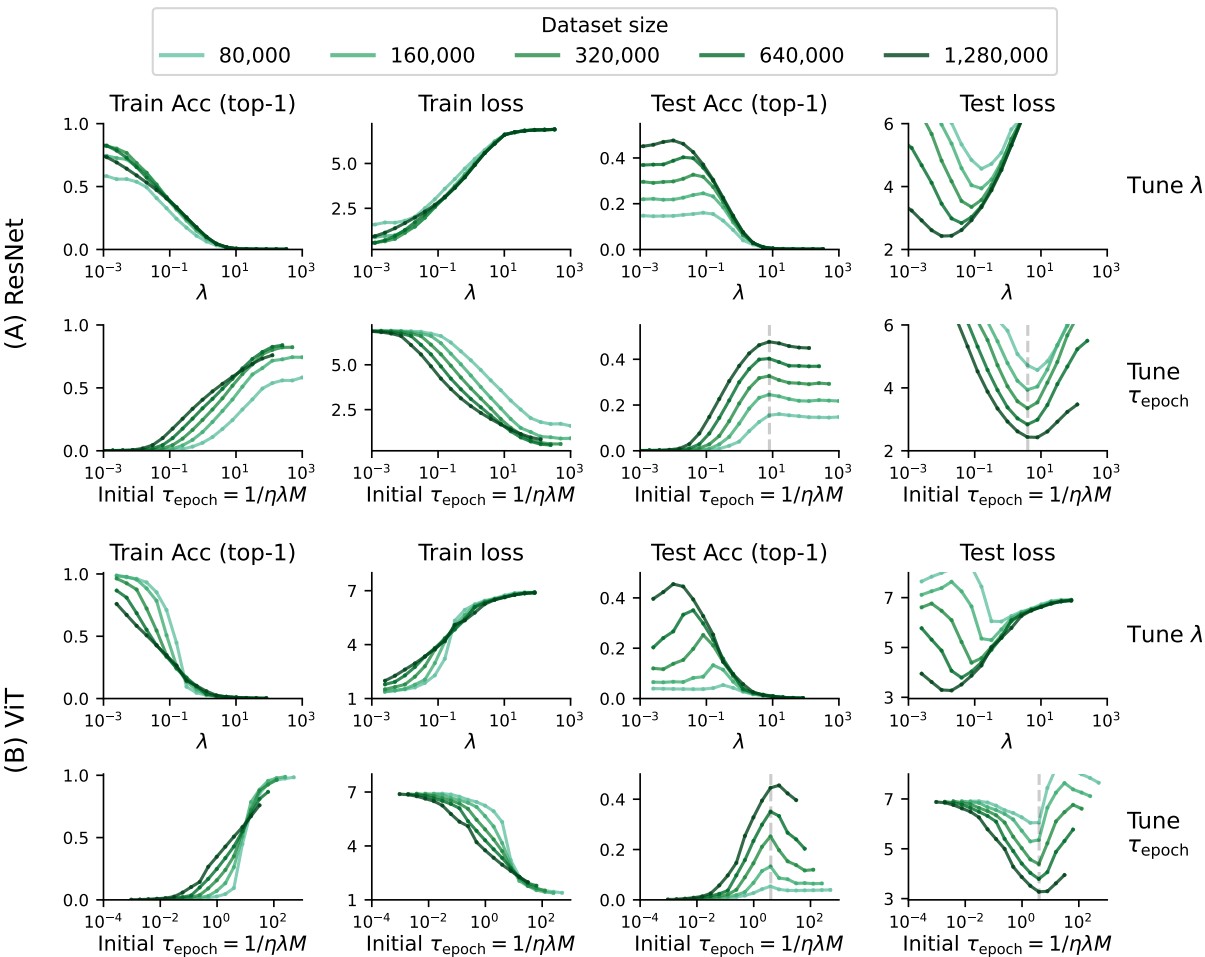

Figure 13: **Optimal $\lambda$ and $\tau_{\text{epoch}}$ v.s. dataset sizes under a constant learning rate.** Similar to Fig. 1 but use a constant learning rate of $10^{-3}$ rather than cosine decay to 0.1 of the initial learning rate.

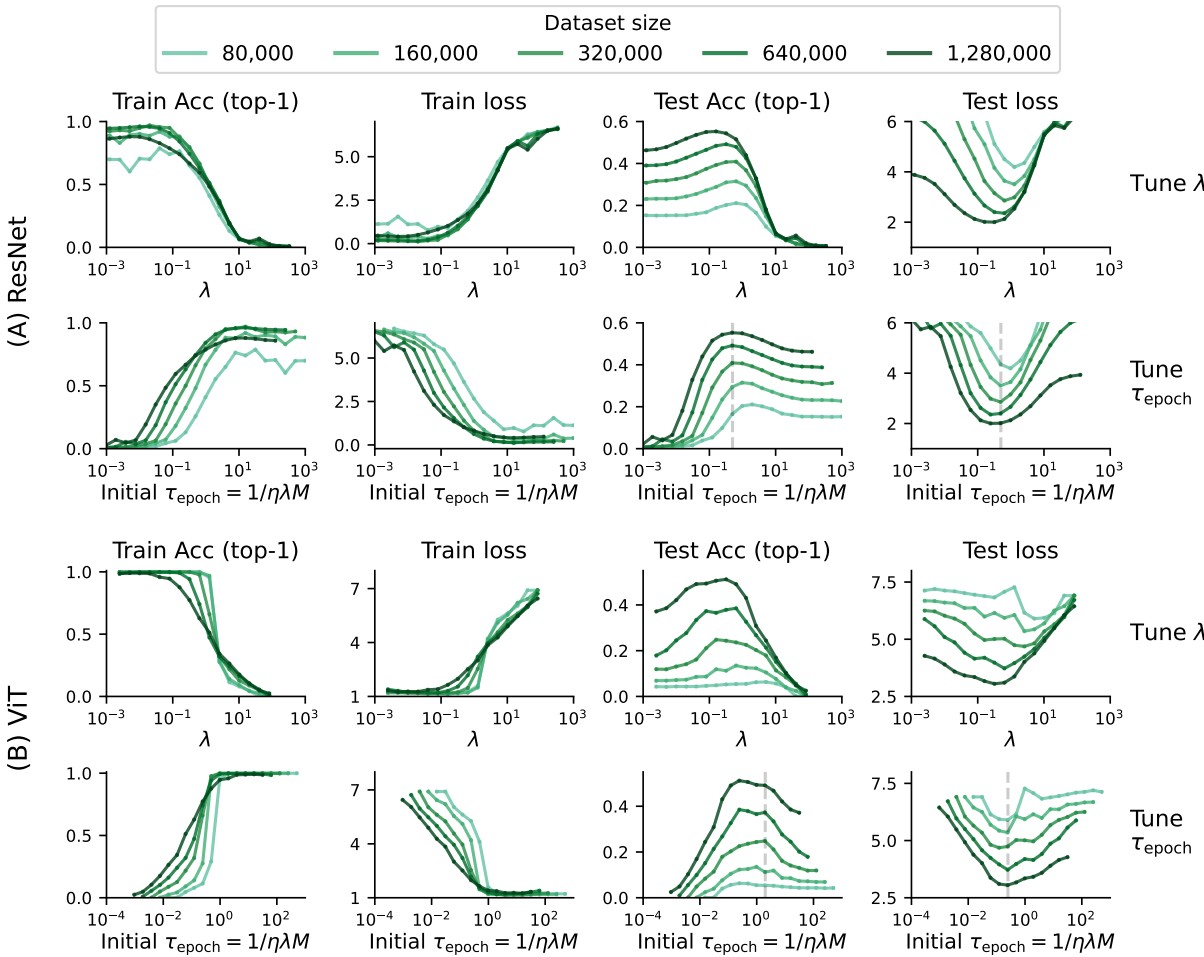

Figure 14: **Optimal $\lambda$ and $\tau_{\text{epoch}}$ v.s. dataset sizes under a cosine decay schedule to 0.** Similar to Fig. 1 but use a cosine learning rate decay schedule from $10^{-3}$ to $o$ rather than to $10^{-4}$.

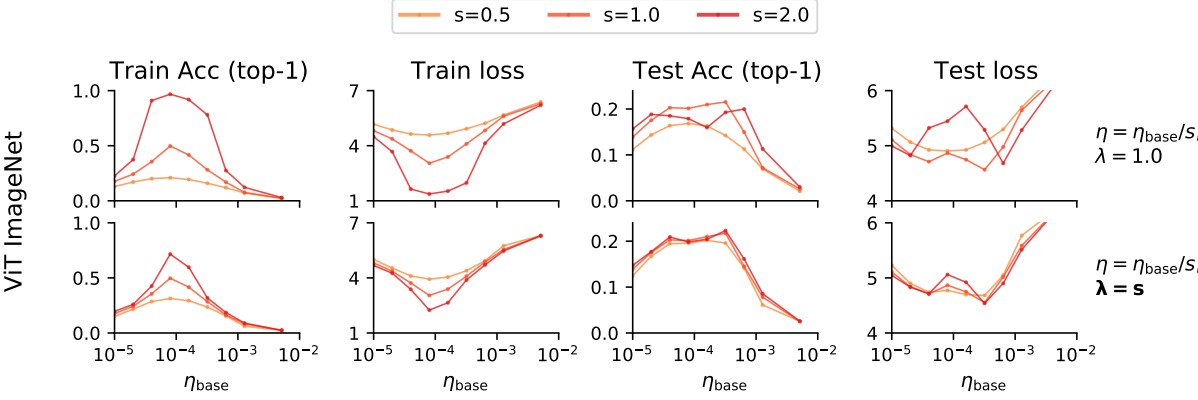

Figure 15: **AdamW breaks the learning rate scaling of μP on ViT.** Similar to the setting in Fig. 5, here we trained a ViT with different width factors on the 320K subset of ImageNet 32x32 under the direct μP scaling (Eq. 13; top row) and our proposed scaling (Eq. 15; bottom row). The direct scaling breaks the transferability of optimal $\eta_{\text{base}}$ due to changing the timescale, whereas our scaling allows stable optimal $\eta_{\text{base}}$ across model sizes.

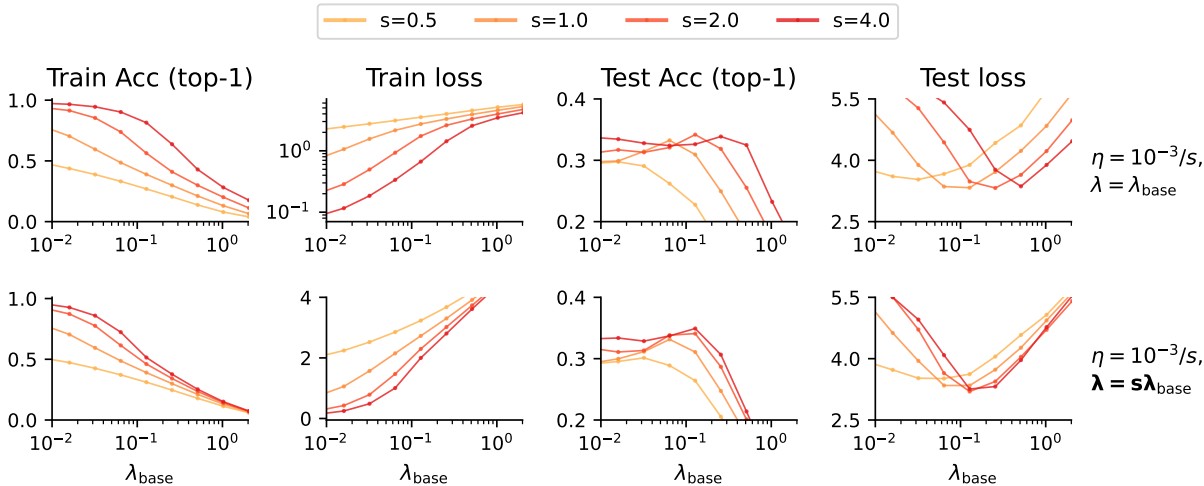

Figure 16: Similar setting to Fig. 3, but use **constant learning rate**.

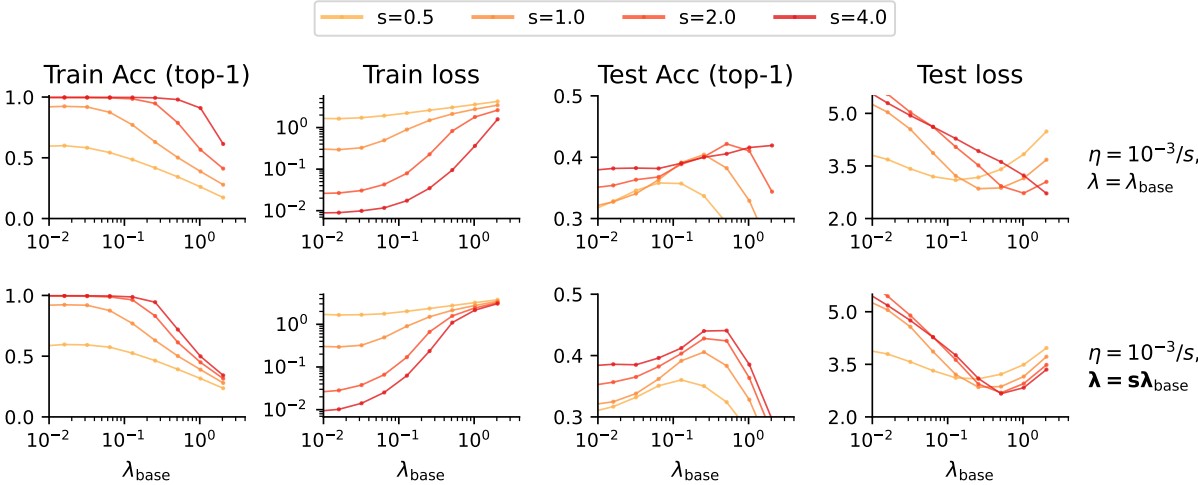

Figure 17: Similar setting to Fig. 3, but use **cosine decay to zero**.

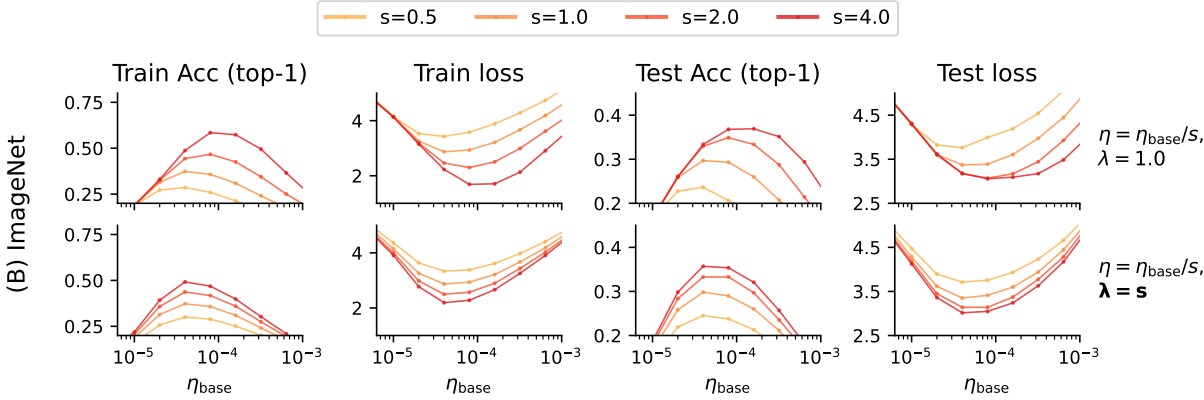

Figure 18: Similar setting to Fig. 5, but use **constant** learning rate.

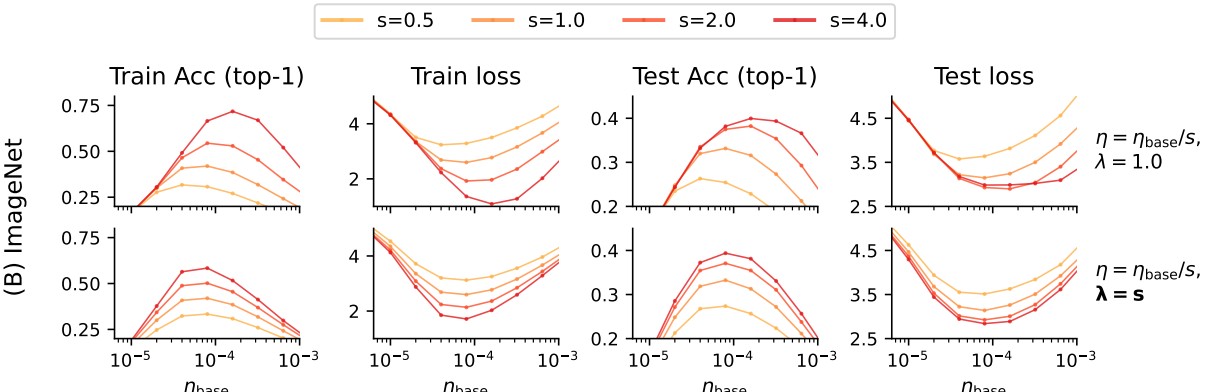

Figure 19: Similar setting to Fig. 5, but use **cosine decay to zero**.

