# OpenReview forum: "How to set AdamW's weight decay as you scale model and dataset size"
_ICML.cc/2025/Conference — ICML 2025 poster_

### Official Review · Reviewer_xMai · 2025-03-08

**Overall Recommendation:** 4

**Summary:**

This paper proposes a simple framework for understanding weight decay in AdamW through its similarity to an exponential-moving-average of weight updates. The hypothesize that the timescale of the EMA (analogous to the half life) is a useful quantifier of the training dynamics. In particular, the authors show that the timescale should be preserved when scaling either the model width or dataset size. This is empirically validated by experiments across several (smallish) deep learning tasks. This offers useful guidance on how to scale the weight decay hyperparameter across task sizes, which is of significant value to the community.

**Post-rebuttal update:** The authors have promised changes that sufficiently address all my concerns. I have raised my score from 3 (weak accept) to 4 (accept).

**Claims And Evidence:**

In general the claims made in the paper are decently supported by evidence. Below are some (minor) comments about the exact phrasing.

**The weights generated by AdamW can be written as an EMA of recent weight updates:** Typically EMA is viewed as a smoothing operation (e.g. in the wikipedia reference provided in the paper). In this case the input sequence typically doesn't depend on the EMA output. This is not the case in AdamW, later updates depend on the previous ones. With a typical EMA in signal processing, you would expect different values of the time constant to vary the amount of smoothing. In AdamW it can give you an entirely different input sequence as well (fundamentally different, more like a dynamical system than EMA). To derive Equation 9, there is also the key assumption of the initial weights being zero (almost never the case in deep learning). Finally the learning rate changing over time would skew the exact EMA form. Overall I feel it would be more fair to say that it can be **approximated as an EMA** where these core assumptions or simplifications are prominently stated.

**The optimal weight decay scales with 1/dataset size:** I think this is conditioned on the learning rate being constant. There are some other works that claim the learning rate should be lower for longer runs (see Schaipp 2501.18965 for a recent example). If that modification is made I am not sure the optimal weight decay is lower. The paper focuses on showing the EMA time coefficient for longer runs is lower, but there are two avenues for lowering this (via either $\eta$ or $\lambda$).

**The optimal weight decay increases with model width:** Similar to the previous point, I believe this assumes the learning rate is scaled via muP. If the learning rate is kept constant (no muP) the optimal weight decay may stay the same.

**Essential References Not Discussed:**

Yes, I feel that key components of this work are closely related or overlap with prior literature on effective learning rates (see above), but this is not sufficiently discussed.

**Experimental Designs Or Analyses:**

I reviewed the experiments and they seem reasonable. It would be nice to see experiments on larger scale networks or datasets, but I understand this may not be feasible.

**Methods And Evaluation Criteria:**

The proposed framework of viewing weight decay as an approximates EMA makes sense and the scaling rules are appropriate and useful for deep learning.

**Other Comments Or Suggestions:**

I think characterizing (Blake 2024) and (Bjorck 2024) as "follow-up works that follow up on the concepts presented in this paper and cite us" is a bit of a stretch (they seem more like concurrent works).

You use "swiped" in several places where I think you might mean "swept".

The paper could use a thorough read through for spelling mistakes.

**Other Strengths And Weaknesses:**

The main strength of the paper is providing a simple view on weight decay and practical guidance on how to set it backed by sufficient although small scale experiments. The main issues are undisclosed relations to prior work and sometimes a lack of clarity or specificity. I think these can be easily addressed which I have already reflected in my rating.

The discussion in the paper feels somewhat limited. In particular there are important questions that are not brought up or addressed:
- The paper conjectures that the EMA time horizon should be kept constant but offers little to no explanation for why this should be the case.
- With a learning rate schedule the time coefficient changes throughout training. Is this important?

**Questions For Authors:**

Q1: Do you agree with my characterization of the relationship between the proposed EMA view and relative learning rates or do you know of a better way to explain this relation?

Q2: Which parts of this paper are specific to AdamW and would not apply to SGD with momentum, Lion or other optimizers?

Q3: How would you justify the time coefficient remaining constant during scaling?

Q4: Does the fact that the time coefficient varies throughout training matter and which value should be transferred?

Q5: Is a lower weight decay strictly better for longer runs / larger datasets or is decreasing the learning rate as good?

I would consider raising my score if the authors can convincingly answer these questions and address the other concerns I described before about clarity and relation to prior work.

**Relation To Broader Scientific Literature:**

The paper makes a decent attempt at discussing related concepts in literature. However there are several places where I feel the connections are not sufficiently detailed.

**Using decoupled weight decay in muP**: This paper proposes scaling the weight decay inversely with the learning rate in PyTorch AdamW. This is equivalent to keeping the weight decay constant in the original “fully-decoupled” AdamW. Other works found this empirically before this paper (e.g. Wortsman 2309.14322). I feel this should be made more explicit in the paper, especially since one of the highlighted claimed contributions is showing “When using AdamW with fixed weight decay, μP learning rate scaling breaks down, but proper weight decay scaling restores its effectiveness”. I believe this exact conclusion could be drawn from prior work as well.

**Effective learning rates:** Viewing AdamW as an update-EMA is **very closely related** to using “relative” (or “rotational”) effective learning rates (Wan 2006.08419, Kosson 2305.17212). The paper briefly mentions these works but does not explain the relation (which is crucial for understanding how this work relates to a large body of prior work on weight decay). The relative learning rate specifies how large the relative change in the weights is on average. Intuitively it shouldn't be too hard to see why this is conceptually related to an EMA, decaying the prior weights at each timestep by e.g. 10% may result in something like a 10% relative change in the weights on average. This can also be formally shown. Given some conditions on the input sequence which are also needed for most of the EMA interpretations to make sense, there is a direct mapping from the EMA decay coefficient to the relative learning rate in the steady-state (see brief sketch below). The other way also works but is less specific. EMA is one way of achieving a given relative learning rate, but there are others e.g. for scale-invariant systems you can exponentially increase your learning rate schedule instead of doing EMA (Li 1910.07454) or keep the weight norm constant and modify the update size accordingly (Kosson 2305.17212).

EMA to Relative Updates: Consider an EMA given by $a_{t+1} = (1-\gamma)a_t + \gamma b_t$. Let's assume that the $b_t$ values are independently random and identically distributed over time with mean zero, and focus on the relative updates in the steady state (large $t$). The relative update size for this scalar case is best defined as $r = \sqrt{\mathbb{E}[(a_{t+1}-a_{t})^2] / \mathbb{E}[a_t^2]}$. To compute $r$ we need to compute $\mathbb{E}[(a_{t+1}-a_t)^2] = \gamma^2 (\mathbb{E}[b_t^2] + \mathbb{E}[a_t^2])$, using independence and mean zero. We can relate $\mathbb{E}[a_t^2]$ to $\mathbb{E}[b_t^2]$ by expanding $a_t$ in terms of $b_t$, squaring, removing the cross terms that are zero due to independence, and finally approximating the sum as infinite (large $t$). This gives $\mathbb{E}[a_t^2] = \frac{\gamma^2}{\gamma^2 - 2\gamma} \mathbb{E}[b_t^2]$, which results in $r=\sqrt{2\gamma}$. Plugging this in **gives an exact match between the EMA timescale proposed in this work and the prior work on effective learning rates in AdamW** (Kosson 2305.17212). Overall I feel the EMA timescale, EMA decay rate, and the relative learning rate are essentially just slightly different characterizations of the same underlying phenomenon. The time coefficient may be a useful quantity as the authors argue, but it feels more like a slight variation of existing works rather than a fundamentally new perspective on weight decay. Like the time coefficient, the effective learning rate also has its strengths. For example it is better defined when the learning rate is changing, it can be measured directly, it allows you to obtain the same learning dynamics without weight decay and / or eliminate the arbitrariness in the weight norms, and is useful for transferring hyperparameters from one optimizer to another.

**Similar or identical conclusions to prior work:** Some of the phenomena discussed when explaining how "EMA view of AdamW provides an explanation for several striking observations made in the literature" have already been explained by prior work, for example Kosson 2305.17212. These include identifying two separate processes for the learning rate and weight decay (lines 387L to 398L) and the scheduling effects where $\eta$ matters at the beginning but $\eta\lambda$ later in training (line 402L). Note that as discussed by Kosson 2305.17212, changing the global learning rate while keeping the weight decay fixed also affects the learning dynamics of gains and biases (i.e. 3 effects not just the 2 mentioned), something that matters in practice as observed in Appendix A but is not discussed around lines 376R and 393L.

**Theoretical Claims:**

I read through them but did not verify the math step by step.

---

> ### Author Rebuttal · Authors · 2025-03-28
>
> Thanks for your extraordinarily extensive and thoughtful review! We agree with all your points and have extensively updated our working manuscript in response.
>
> ## Claims And Evidence
>
> **Approximate EMA:** Fixed.
>
> We do NOT **assume that the initial weights are zero in Eq 9**!  We assume that the initializations for the two networks use the same random noise, but different scalings (Eq 8).
>
> We are assuming **learning rate rate does not change as we train on more data** (see 375r-379r, which mirrors a point from Kosson), and we assume **muP scaling for width**.
>
>  We have (re) emphasized all these points in the manuscript.
>
> ### Q1 Prior work
>
> Thanks for highlighting **Kosson**: This is a great paper that should have had a bigger impact.
> We cite Kosson in the related work (line 424l), stating that the key difference with our work is that Kosson does not discuss hyperparameter transfer.
> Nonetheless, we agree that there are more connections and have extended the discussion of Kosson in the Related Work, along with writing an "extended prior work" section that fully elaborates on this connection.
> Let us know if you'd like specific wording as we frame this connection!
>
> **Wortsman** studies the relationship between the final performance and learning rate.
> In their Fig 6, they find that performance is insensitive to learning rate, $\eta$, when fixing $\eta \lambda$, while it is more sensitive if $\lambda$ is fixed.
> We see this as directly connecting to the **Kosson** result, that $\eta$ matters early on and $\eta \lambda$ matters later on.
> The connection to our contributions on EMA and hyperparameter transfer are *definitely there*, but are far more indirect (though let us know if you're actually thinking of a different result from Wortsman).
>
> ### Q2
> We believe the result is **more general than AdamW**, applying to any setting where the updates have roughly constant magnitude as the scale of the underlying parameters varies, eg Lion Sophia Muon etc but not SGD.
>
> ### Q3: EMA timescale constant
>
> The EMA viewpoint suggests a reason why the time horizon should be constant. We have edited the manuscript to include this fuller discussion.
>
> An EMA can very approximately be understood as averaging over the last $\tau_{iter}$ samples (Eq 7), as it downweights considerably any datapoints more than $\tau_{iter}$ steps ago.
>
> What does this mean for training neural networks?
>
> We assume that each datapoint provides useful information, so you can't drop or considerably downweight datapoints without harming performance. As discussed above, the EMA downweights considerably datapoints that it saw more than $\tau_{iter}$ iterations ago. Thus, to avoid considerably downweighting datapoints, $\tau_{iter}$ should be larger than the number of iterations required for all the datapoints (i.e. one epoch).
>
> The above reasoning suggests that you can set $\tau_{iter} = \infty$, or $\lambda = 0$, as that would would average over every update. But it would also reduce AdamW to Adam.
>
> Of course, we know that AdamW (i.e non-zero $\lambda$) works better.
> To resolve this conundrum, we conjecture that you don't want to average over all updates. We believe that updates from very early in training are detrimental, as they are based on early settings of the weights. To forget these initial updates we can use a $\tau_{iter}$ which is smaller than the total number of iterations in the training run.
>
> These two observations give a "natural range" for the optimal $\tau_{iter}$: somewhere between the number of iterations for one epoch and the total number of iterations.
> This natural range is simpler if you measure the timescale in terms of epochs: it's somewhere between $1$ and the total number of epochs.
> We find experimentally that the optimal timescale does lie in this range (Fig 1,2).
> As this natural range is fixed wrt model and dataset size, we conjectured that the optimal EMA timescale was fixed.
>
>
> ### Q4
>
> We transfer the sequence of timescales through training.
>
> We believe it's optimal to have a scheduled timescale across training: scheduling makes it easier to forget detrimental initial updates while doing long-timescale averaging towards the end of training.
>
> All existing "decoupled weight decay" implementations (e.g Composer from MosaicML) change the timescale by changing the learning rate. While you could also use $\lambda$, we leave that for future work.
>
> ### Q5
> We agree that you could **modify timescale using either $\eta$ or $\lambda$**. Our motivation for modifying $\lambda$ is explained in the first paragraph of the Related work (line 375r-379r).
>
> Related to this, we ran an extra [experiment](https://anonymous.4open.science/r/3MDG98-66F3/response_pdf.pdf) showing that if you fix $\lambda$, then the optimal $\eta$ changes as dataset size increases.  In contrast, if you use our scaling for $\lambda$, then the optimal $\eta$ is fixed.  We will run a complete set of experiments on this for the camera-ready.

---

> > ### Comment · Reviewer_xMai · 2025-04-01
> >
> > Thank you for thoroughly addressing my concerns. I have no significant concerns remaining and will raise my review score to accept to reflect this.
> >
> > Some brief responses / acknowledgements:
> > - Equation 9: Yes, I meant Equation 7 for the EMA update, sorry.
> > - On Wortsman: Agreed, the paper focuses more on learning rate sensitivity than transfer. One of the practical takeaways for me was to use decoupled weight decay for muP, but it doesn't really justify it as muP breaking down rather than increased learning rate sensitivity. I think this is sufficiently different to justify your claim.
> > - Thank you for addressing the rest of the concerns and answering the questions, they all seem good to me.

---

> > > ### Author Response · Authors · 2025-04-01
> > >
> > > Dear reviewer xMai
> > >
> > > Thank you for raising the score, and we sincerely thank you again for your insightful, detailed, and extensive comments. With the suggested changes and clarifications included, we indeed find the manuscript to have much better clarity and specificity!
> > >
> > > Thanks,
> > >
> > > Authors

---

### Official Review · Reviewer_YYQ3 · 2025-03-13

**Overall Recommendation:** 2

**Summary:**

This paper studies the AdamW optimizer. the authors provide empirical studies on the hyperparameters of the AdamW under different settings. Specifically, the paper first reformulate the AdamW itself is an EMA. then provide experiments on resnet, vit and LLM training, showing some empirical weight decay parameters tuning rules when increasing the dataset and model size. Finally, the authors discussed the relationship between weight decay and \muP learning.

**Claims And Evidence:**

yes

**Essential References Not Discussed:**

None

**Experimental Designs Or Analyses:**

yes

**Methods And Evaluation Criteria:**

yes

**Other Comments Or Suggestions:**

none

**Other Strengths And Weaknesses:**

Strength
- the authors conducted extensive numerical experiments to support their claims.
- the reported results could be helpful in ML applications.


Weaknesses:
- the paper is mainly empirical results based. though the authors provided experiments on resnet, LLM, vit, it could be difficult to generalize to all other machine learning models.
- the reformulation of AdamW to EMA looks trivial and incremental to me. the author didn't claim what are the benefits by this reformulation.
- though there are extensive experimental evidence, the theory part of this paper is weak. the authors didn't provide solid theoretical understanding to the phenomenon they observed.

**Questions For Authors:**

none

**Relation To Broader Scientific Literature:**

None

**Theoretical Claims:**

yes

---

> ### Author Rebuttal · Authors · 2025-03-27
>
> Thanks for your careful review.
>
> # Value of the EMA perspective
>
> We fully agree that the reformulation of AdamW to EMA is almost trivial.
> Our key insight was noticing that this almost trivial connection provides novel, powerful insights into hyperparameter transfer for weight decay, that we then validated extensively.
> It is precisely the simplicity of our approach that makes it easy for people actually training neural networks to understand and apply.
> Indeed, we know that at least one "unicorn" is using training recipes inspired by the work here for training large scale foundation models.
>
> Finally, we believe our work is **the very first paper** studying the problem of transferring weight decay parameters across problem sizes.
>
> # Experiment design
>
> As you noted, we took the rules for hyperparameter transfer suggested by the EMA view, and validated them through extensive experiments on the three major classes of machine learning models (ResNets, LLMs and VITs).
>
> # Theoretical understanding
>
> While our results are supported by Theorem 1 (proof in Appendix A1), we do agree that our results are mainly around validating the simple but powerful viewpoint on hyperparameter transfer provided by the connection between AdamW and EMA.

---

### Official Review · Reviewer_9CyY · 2025-03-13

**Overall Recommendation:** 5

**Summary:**

The authors study the scaling behavior of the optimal AdamW weight decay hyperparameter with respect to model and dataset sizes. They provide a theoretical insight by framing AdamW's learned weights as an exponential moving average (EMA) of recent updates, identifying the EMA timescale as the key underlying hyperparameter. Empirically, the authors demonstrate that the EMA timescale (in epochs) remains roughly constant across scales.

## update after rebuttal
I confirm that I have read the authors' response to my review and have revised my review accordingly, where appropriate.

**Claims And Evidence:**

In theorem 1, is the "scale-invariant network" assumption too strong?

**Essential References Not Discussed:**

The primary contribution is identifying and characterizing the scaling rule for weight decay. While 'Implicit Bias of AdamW: ℓ∞ Norm Constrained Optimization' illustrates the implicit bias introduced by weight decay, earlier work such as 'Lion Secretly Solves Constrained Optimization: As Lyapunov Predicts' establishes the connection between the norm of trained neural network parameters and the weight decay coefficient λ, demonstrating this relationship for both Lion and AdamW.

**Experimental Designs Or Analyses:**

This leads to a direct rule for scaling AdamW weight decay: the optimal weight decay decreases with increasing dataset size and increases with model size (under the muP learning rate scaling recommendations).

**Methods And Evaluation Criteria:**

Most experiments are conducted on vision tasks, but they have multiply epochs, while for LLM pre-training, we only do single epoch given the large data size.

It would be great if the author could give a table to summarize the scaling "rules" of weight decay in terms of model size, data size, batch size, etc.

**Other Comments Or Suggestions:**

EMA of update can be seem as kind of nice perspective to see weight decay. Have the authors use continuous time

**Other Strengths And Weaknesses:**

Novelty and Impact:

The topic is relevant and timely, addressing an important practical challenge in the era of large-scale neural networks. The paper contributes substantially to our understanding of hyperparameter transferability across model and data scales.

**Questions For Authors:**

Have you explored extending these insights to other adaptive optimizers, such as AdaGrad or AdaFactor, to investigate if similar implicit bias or parameter norm constraints exist?

**Relation To Broader Scientific Literature:**

EMA of update has been well studied, for example in Theorem B.10. of "Lion Secretly Solves Constrained Optimization: As Lyapunov Predicts".

**Theoretical Claims:**

The authors' interpretation of AdamW's weight updates as an exponential moving average (EMA) provides valuable insights. However, similar EMA-based interpretations of adaptive optimizers and their implicit constraints have been recently explored in related literature. In particular, Chen et al. (ICLR 2024, Theorem B.6, "Lion Secretly Solves Constrained Optimization: As Lyapunov Predicts") and Liu et al. (NeurIPS 2024, Theorem A.1, "Communication Efficient Distributed Training with Distributed Lion") explicitly derived EMA forms for optimizer updates, highlighting their implicit constrained optimization behavior. Additionally, Xie and Li (ICML 2024, "Implicit Bias of AdamW: ℓ∞-Norm Constrained Optimization") similarly analyzed AdamW's implicit constraints.

To appropriately contextualize and acknowledge prior work, the authors should clearly cite and discuss these papers, clarifying how their EMA formulation aligns with or differs from these recent studies.

---

> ### Author Rebuttal · Authors · 2025-03-27
>
> Thanks for your extremely positive review!
>
> Great catch with Chen et al. (Theorem B.6) and Liu et al. (Theorem A.1)! We have added a discussion of these papers to our working draft and adjusted our contributions section. In short, we aren't surprised that someone has used an EMA-like result/form as an intermediate result in other calculations.  Our contribution is in noticing that this simple connection between AdamW and EMA provides a powerful lens on **hyperparameter scaling** specifically for weight decay.
>
> We have also added a discussion of Chen et al. (ICLR 2024) and Xie and Li (ICML 2024) to the Related Work.  As far as we can see, the "implicit constraints of AdamW" noticed in these works work, refers primarily to the relationship between *the scale of the weight* and *weight decay*, which we briefly discussed in Appendix C, Fig 11, but did not claim as a contribution.  To our understanding, this work did not study the transfer/scaling of weight decay across problem sizes, which is our primary contribution.  Though let us know if you're thinking of a different result here.
>
> > In theorem 1, is the "scale-invariant network" assumption too strong?
>
> We aren't quite sure what you mean by "too strong" here.  Do you mean that we could prove the same result with a weaker assumption?  While this may be possible, we don't see how.  Intuitively, the need for the scale-invariant network assumption arises from the $1/\lambda$ scaling of the weights in AdamW.  If the scale of the weight matters (i.e. the network is not scale invariant), then $\lambda$ on its own affects the learning trajectory, in addition to the EMA timescale, and the ratio between the learning rate and the initialization scale $\rho = \eta / \sigma$.
>
> > It would be great if the author could give a table to summarize the scaling "rules" of weight decay in terms of model size, data size, batch size, etc.
>
> We have added this table to our working draft.
>
> > Have you explored extending these insights to other adaptive optimizers, such as AdaGrad or AdaFactor, to investigate if similar implicit bias or parameter norm constraints exist?
>
> Not as of yet, but we do expect the same considerations to transfer across all settings with constant magnitude updates and decoupled weight decay, including SignGD, Lion, Sophia, Muon, SOAP etc.  For instance, it was recently noted that Muon with decoupled weight decay has the same $1/\lambda$ scaling of the max eigenvalue (see section Weight Decay from [1])
>
> [1] Why We Chose Muon: Our Chain of Thought? https://x.com/Kimi_Moonshot/status/1897929976948965870

---

### Decision · Program_Chairs · 2025-05-01

**Decision:**

Accept (poster)

**Comment:**

The paper notes a simple connection between EMA and weight decay to motivate a new hyperparameter transfer scheme. It performs thorough experiments to validate the approach. Given the empirical importance of weight decay, the paper's questions, investigation, and results can significantly aid practitioners. Two positive reviewers justify their scores thoroughly while one negative reviewer argues the paper doesn't have enough theoretical contribution, but the paper's main contribution is empirical. The paper can do a better job of contextualizing itself with prior work that implies related statements (e.g., Wortsman et al.), which the authors have promised to do.